# Feeding-dependent tentacle development in the sea anemone *Nematostella vectensis*

Aissam Ikmi [1,2✉], Petrus J. Steenbergen[1], Marie Anzo [1], Mason R. McMullen[2,3], Anniek Stokkermans[1], Lacey R. Ellington[2] & Matthew C. Gibson[2,4]

In cnidarians, axial patterning is not restricted to embryogenesis but continues throughout a prolonged life history filled with unpredictable environmental changes. How this developmental capacity copes with fluctuations of food availability and whether it recapitulates embryonic mechanisms remain poorly understood. Here we utilize the tentacles of the sea anemone *Nematostella vectensis* as an experimental paradigm for developmental patterning across distinct life history stages. By analyzing over 1000 growing polyps, we find that tentacle progression is stereotyped and occurs in a feeding-dependent manner. Using a combination of genetic, cellular and molecular approaches, we demonstrate that the crosstalk between Target of Rapamycin (TOR) and *Fibroblast growth factor receptor b* (*Fgfrb*) signaling in ring muscles defines tentacle primordia in fed polyps. Interestingly, *Fgfrb*-dependent polarized growth is observed in polyp but not embryonic tentacle primordia. These findings show an unexpected plasticity of tentacle development, and link post-embryonic body patterning with food availability.

[1] Developmental Biology Unit, European Molecular Biology Laboratory, 69117 Heidelberg, Germany. [2] Stowers Institute for Medical Research, Kansas City, MO 64110, USA. [3] Department of Pharmacy, The University of Kansas Health System, Kansas City, KS 66160, USA. [4] Department of Anatomy and Cell Biology, The University of Kansas School of Medicine, Kansas City, KS 66160, USA. ✉email: aissam.ikmi@embl.de

Cnidarians such as sea anemones, corals, and hydrozoans have continuous developmental capacities[1–3]. Similar to plants, these early-branching animals can generate organs and body axes throughout their entire life. This developmental feature underlies diverse phenomena, such as secondary outgrowth formation[4], asexual reproduction[5], and branching in colonial species[6]. The ability to continuously build new body parts is comparable to regeneration, as both require activation of patterning mechanisms in a differentiated body plan. However, unlike regeneration induced by damage or injury, life-long organogenesis is subject to environmental modulation. This strategy allows cnidarians, like plants, to continuously adjust their developmental patterns to unpredictable fluctuations of food supply[2,7]. To determine how cnidarians regulate organogenesis across life history stages, and whether this process recapitulates embryonic development or employs distinct mechanisms, here we study postembryonic tentacle development in the starlet sea anemone Nematostella vectensis.

Arrays of tentacles armed with stinging cells are a unifying feature of Cnidaria, with diverse species featuring distinct arrangements, morphologies, and numbers of tentacles[8–11]. In the typical cnidarian polyp bauplan, oral tentacles are simple extensions of the diploblastic body, forming appendages that feed, defend, and expand the surface area of the gastric cavity. Zoologists in the early 1900s described tentacle patterns in select species, some of that could exceed 700 tentacles (e.g., Cereus pedunculatus)[10]. The partial sequence by which tentacles are added overtime was also reported, reflecting the existence of continuous axial patterning in polyps. In hydrozoans, this developmental property was primarily studied in the context of body plan maintenance and regeneration, where the oral tissue of Hydra features a Wnt/ß-catenin-dependent axial organizing capacity[12]. Still, how new morphological patterns are generated and how developmental patterning unfolds across distinct life history stages remain unknown. Interestingly, a link between tentacle morphogenesis and nutrition was recently observed in the sea anemone Aiptasia[13], but the mechanistic basis of this nutrient dependency is still unsolved.

In recent years, N. vectensis has become an established cnidarian model in developmental biology due to its relative ease of laboratory spawning[14,15], tractable developmental biology[16–20], extensive regenerative capacity[21–24], and robust molecular genetic approaches[25–29]. Nematostella polyps can harbor a variable number of tentacles ranging from 4 to 18, but the common number in adulthood is 16 (refs. [4,10]). During the development, four-tentacle buds simultaneously form in the swimming larvae and give rise to the initial appendages of the primary polyp[4]. The formation of tentacles involves coordination between both embryonic body layers. In developing larvae, an endodermal Hox code controls inner-body segmentation, defining territories that are critical for tentacle patterning[20,30,31]. In the ectoderm, tentacle morphogenesis initiates from a thickened placode, followed by changes in epithelial cell shape and arrangement that drive tentacle elongation[4]. Nevertheless, we still have only a rudimentary understanding of the developmental relationship between embryonic and postembryonic tentacles, or how tentacle progression integrates the nutritional status of the environment.

In this study, we analyze over 1000 growing Nematostella polyps, and identify a stereotyped pattern of tentacle addition that is feeding-dependent. Using in situ hybridization and a transgenic reporter line, we show that discrete Fgfrb-positive ring muscles prefigure the sites of new tentacles in unfed polyps. In response to feeding, a TOR-dependent mechanism controls the expansion of Fgfrb expression in oral tissues that defines tentacle primordia. By generating a knockout line, we demonstrate that Fgfrb is required to regionally enhance TOR signaling activity and promote polarized growth, a spatial pattern that is restricted to polyp, but not to embryonic tentacle primordia. These results identify distinct trajectories of tentacle development, and show that the cross talk between TOR-mediated nutrient signaling and FGFRb pathway couples postembryonic body patterning with food availability.

## Results

**Tentacle patterning in primary and adult polyps**. Nematostella polyps possess two axes: one running from the pharynx to the foot (oral–aboral axis), and an orthogonal axis traversing the pharynx (directive axis, Fig. 1a, b). Along the directive axis, the primary polyp displays mesenteries, internal anatomical structures that subdivide the body into eight recognizable radial segments (s1–s8; Fig. 1b and Supplementary Fig. 1)[20]. The four primary tentacles occupy stereotyped radial positions corresponding to segments s2, s4, s6, and s8 ($n = 131$ polyps, Fig. 1b). While this octo-radial body plan is maintained in adults[32], we found that the spatial pattern of tentacle addition generated three new features in mature polyps with 16-tentacles (Fig. 1b and Supplementary Fig. 1). First, tentacles were arranged in a zigzag pattern forming two adjacent concentric crowns at the oral pole. Second, the boundaries between neighboring tentacles within the same segment were marked by the formation of short gastrodermal folds, enriched in F-actin and called microcnemes (Supplementary Fig. 1)[10]. Third, the arrangement of tentacles shifted from radial to bilateral organization with a defined number of tentacles in each segment. This included a single tentacle in the directive segments s1 and s5, two tentacles in segments s2, s8, s3, and s7, and three tentacles in segments s4 and s6 ($n = 108$ polyps, Fig. 1b, c). Together, these observations reveal that tentacle development in polyps derives from an intricate and reproducible spatial patterning system. In addition, we observed that adult polyps can grow >18-tentacles when they were not regularly spawned (Supplementary Fig. 2). Whether wild polyps also exhibit this developmental behavior is unknown, but this finding suggests the existence of a trade-off between resource allocation for reproduction and tentacle development.

**Feeding-dependent stereotyped tentacle formation**. In the absence of nutrients, we found that Nematostella polyps arrested at the four-tentacle stage and rarely developed additional tentacles (Supplementary Fig. 3). When food was available, however, primary polyps grew and sequentially initiated new tentacles in a nutrient-dependent manner, arresting at specific tentacle stages in response to food depletion (Supplementary Fig. 3). To build a spatio-temporal map of tentacle addition, we leveraged this feeding-dependent development to control the progression of tentacle stages. We examined 1102 fixed specimens collected between the 4- and 16-tentacle stages, and recorded the placement and order of tentacle formation (Figs. 2 and 3). Despite using a non-isogenic strain of Nematostella, tentacle addition was surprisingly stereotyped and was dependent on two distinct budding modalities, which we term cis- and trans-budding. In both modes, tentacles were generated in pairs, either through simultaneous or consecutive budding events. In trans-budding, new tentacles formed in opposing segments along the directive axis (Fig. 2a–c). In cis-budding, a pair of new tentacles developed in the same segment (Fig. 3a, b). These budding patterns occurred in a distinct temporal sequence. Based on the spatio-temporal deployment of new tentacles, we conclude that the nutrient-dependent development of tentacles falls into three phases mediated by six pairs of budding events (Fig. 1c).

Phase I corresponded to the progression from the four- to the eight-tentacle stage, where two trans-budding events took place in

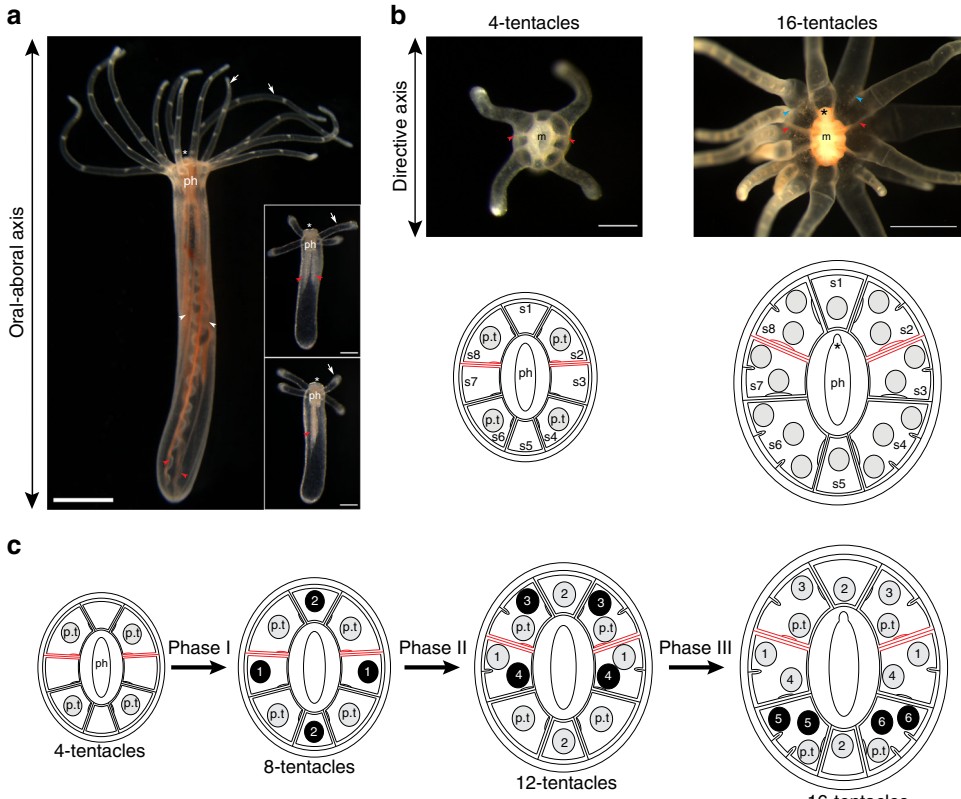

**Fig. 1 Body axes and tentacle arrangement in primary and adult polyps. a** Lateral views of an adult polyp bearing 12 tentacles (left panel, $n = 14$ polyps) and a 4-tentacle primary polyp (right panels, $n = 20$ polyps). Animals were pictured live in three independent experiments. White asterisk (*) indicates the position of the oral pole. The pharynx (ph) is attached to the body wall through eight endodermal mesenteries. Right panels show 90° rotation of a four-tentacle polyp. Two primary mesenteries (red arrowheads) form a mirror image along the directive axis. In adults, these primary mesenteries maintain a growth advantage over the remaining mesenteries (white arrowheads). Note that the position of primary mesenteries can serve as a landmark to orient the oral pole along the directive axis. White arrows show tentacles. Scale bars are 1 mm and 100 μm in left and right panels, respectively. **b** Oral views of 4-tentacle and 16-tentacle polyps ($n = 131$ primary polyps, $n = 108$ adult polyps, three independent experiments). Both oral poles show eight segments and are oriented along the directive axis using the position of primary mesenteries (red arrowheads). Black asterisk (*) indicates the position of the siphonoglyph at one end of the mouth (m). Blue arrowheads indicate two examples of boundaries between neighboring tentacles within segments. Scale bars are 250 μm and 1 mm in left and right panels, respectively. Lower panels: diagrammatic cross-section through the oral pole summarizes the arrangement of tentacles. The eight body segments are annotated from segment 1 (s1) to segment 8 (s8). Primary mesenteries are colored in red. Tentacles are depicted as gray discs. Primary tentacles (p.t) resulting from embryonic development are indicated. **c** Summary of the representative pattern of tentacle addition showing the three phases. The sequence of budding events is indicated as black discs with a number.

the tentacle-less segments s1, s3, s5, and s7 (Fig. 2a, b, e). Tentacle buds initially developed in segments s3 and s7, followed by segments s1 and s5. The first *trans*-budding event was mostly simultaneous, while the second showed many cases of asynchrony (Fig. 2b, e). At the end of phase I, each of the eight segments displayed a single tentacle. In phase II, a bilateral pattern of tentacles emerged as a result of two *trans*-budding events leading to 12-tentacle stage (Fig. 2c–e). The third and fourth pairs of tentacles were added in segments s2/s8 and s3/s7, respectively. In contrast to phase I, *trans*-budding events were preceded by the formation of short gastrodermal folds (Supplementary Fig. 1). These new boundaries created territories for tentacle development within segments, with buds generated in stereotyped locations with respect to the preexisting tentacles. During phase III, the tentacle development relied on the *cis*-budding mode to proceed from the 12- to 16-tentacle stage (Fig. 3a–c). The fifth and sixth pairs of tentacles were sequentially formed in segments s4 and s6, with no preferential bias in the order of deployment (Fig. 3d).

Although the tentacle addition pattern was highly stereotyped, we also observed some temporal and spatial variability in tentacle pattern (Supplementary Fig. 4). A strong asynchrony in the

development of tentacle pairs was the predominant variant observed during the progression from 4- to 12-tentacle stages ($n = 155/824$). We also found animals where one sister pair of tentacles was skipped or formed in a neighboring segment. In other cases, the order of budding events was inverted or two pairs of bud formed simultaneously ($n = 66$). During phase III, we found a few animals that used *trans*-budding instead of *cis*-budding ($n = 6/320$). Collectively, these variants represented ~20.6% of all specimens scored, perhaps attributable to genetic variation, variation in nutrient uptake, and developmental plasticity in growing polyps.

**Feeding-induced tentacle development is size independent.** To elucidate the mechanisms that control tentacle addition, we focused on the transition from the four- to the six-tentacle stage. Following daily feedings of primary polyps, the first pair of buds appeared between days 4 and 6, depending on the amount of *Artemia* consumed (Supplementary Fig. 3). At the organismal scale, tentacle budding coincided with an approximate twofold increase in body size (Supplementary Fig. 5). To probe the relationship between body size and tentacle budding, we generated reduced-sized primary polyps by dividing four-cell stage embryos

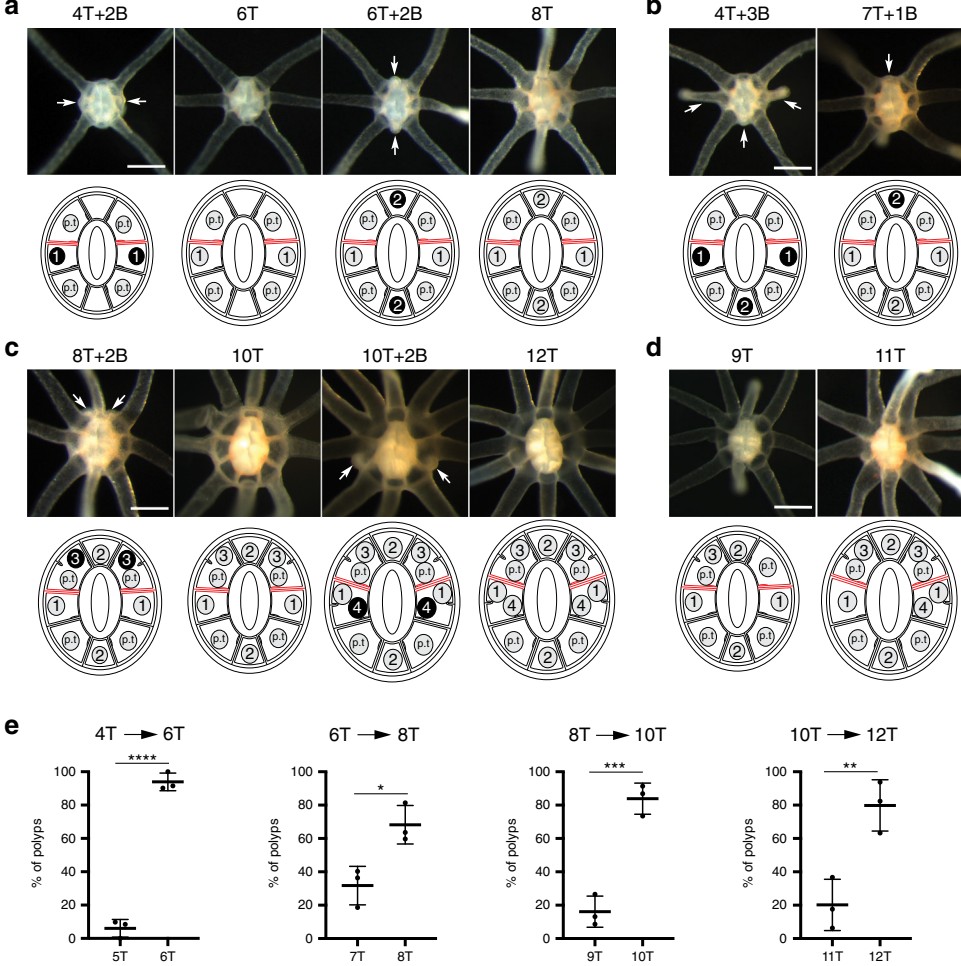

**Fig. 2 Phases I and II of tentacle addition. a**, **b** Oral views of fed polyps progressing from four to eight tentacles. **a** Polyps showing the dominant budding patterns. Polyps with four-tentacles and two buds (4 T + 2B, $n = 64$), six-tentacles (6 T, $n = 88$), six-tentacles and two buds (6 T + 2B, $n = 31$), and eight-tentacles (8 T, $n = 67$). **b** Alternative budding patterns. Polyps with four-tentacles and three buds (4 T + 3B, $n = 17$), and seven-tentacles and one bud (7 T + 1B, $n = 6$). **c**, **d** Oral views of fed polyps progressing from 8 to 12 tentacles. **c** Dominant budding patterns. Polyps with 8-tentacles and two buds (8 T + 2B, $n = 37$), 10-tentacles (10 T, $n = 107$), 10-tentacles with two buds (10 T + 2B, $n = 42$), and 12-tentacles (12 T, $n = 223$). **d** Polyps with 9-tentacles (9 T, $n = 29$) and 11-tentacles (11 T, $n = 69$). White arrows show the sites of budding events. All data are from three independent experiments. Scale bars are 250 μm in all micrographs. Diagram showing tentacle arrangement is provided under each image. Tentacles and buds are depicted as gray and black discs, respectively. The numbers inside the discs indicate the sequence of budding. The positions of primary tentacles (p.t) are shown. **e** Quantification of tentacle number including buds in polyps from three independently growing groups (the number of polyps in each group are $n = 334$, $n = 315$, and $n = 175$). Tentacle progression is indicated on the top of each graph and the numbers of analyzed polyps are indicated. For all the graphs, data are mean ± SD for error bars (unpaired Student's two-tailed $t$-test, $^*p < 0.05$, $^{**}p < 0.01$, $^{***}p < 0.001$, $^{****}p < 0.0001$). Source data are provided as a Source data file.

into two pairs of blastomeres and then allowing development to proceed (Fig. 4a)[33]. Upon feeding, reduced-sized polyps developed tentacle buds at smaller oral circumferences compared to full-sized animals, indicating that the feeding-dependent tentacle development relies on a mechanism that scales in proportion to body size (Fig. 4b, c).

To monitor the growth during tentacle development, we labeled cells in S-phase with EdU incorporation and observed two distinct patterns of cell proliferation. Uniform S-phase labeling was induced in response to feeding. This was followed by a localized increase in cell proliferation in both cell layers of tentacle bud primordia (Fig. 4d and Supplementary Fig. 6). Bud-localized proliferation transformed the thin epithelial layers into a thickened outgrowth, generating the initial cellular organization associated with budding stages (Supplementary Fig. 6). During tentacle elongation, cell proliferation was uniform but excluded from tentacle tips, a region enriched with differentiating stinging cells[34,35]. These results show that tentacle morphogenesis in

primary polyps is preceded by nutrient-dependent global growth. This is in turn followed by the formation of localized growth zones that mark the sites of the nascent buds.

To define the relationship between nutrient input and cell proliferation, we established a minimal feeding assay sufficient to induce tentacle budding. Under these conditions, polyps were fed for 3 days and then starved for 4 days until the first pair of buds developed. Interestingly, while we observed a dramatic reduction of uniform cell proliferation in the starved budded polyps, bud-localized cell proliferation was significantly less affected (Fig. 4e, f). The different sensitivities to starvation suggest that distinct regulatory mechanisms control cell proliferation associated with generalized organismal growth and region-specific tentacle budding. Interestingly, an enrichment of lipid droplets was detected in bud primordia, visualized with BODIPY and Oil Red O staining (Fig. 4g and Supplementary Fig. 7). This localized energy storage could serve as a buffering mechanism to complete tentacle development under unpredictable fluctuations of food supply.

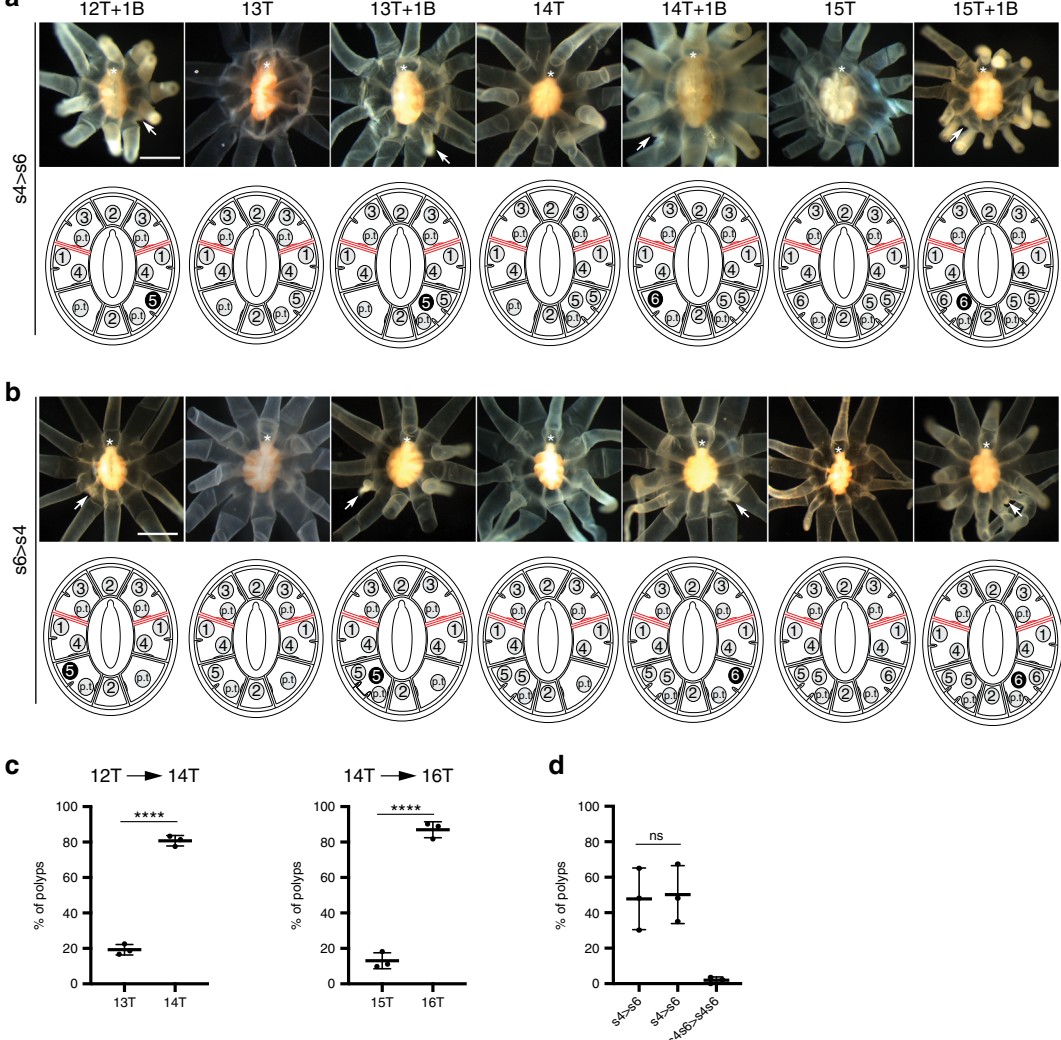

**Fig. 3 Phase III of tentacle addition. a, b** Oral views of fed polyps progressing from 12 to 16 tentacles. **a** *Cis*-budding events taking place in segment 4 then segment 6 (s4 > s6). Polyps with 12-tentacles with one bud (12 T + 1B, $n = 10$), 13-tentacles (13 T, $n = 16$), 13-tentacles with one bud (13 T + 1B, $n = 30$), 14-tentacles (14 T, $n = 43$), 14-tentacles with one bud (14 T + 1B, $n = 3$), 15-tentacles (15 T, $n = 6$), 15-tentacles with one bud (15 T + 1B, $n = 20$). **b** Alternative pattern of *cis*-budding events (s6 > s4, number of polyps are $n = 4$ for 12 T + 1B, $n = 4$ for 13 T, $n = 25$ for 13 T + 1B, $n = 46$ for 14 T, $n = 4$ for 14 T + 1B, $n = 5$ for 15 T, and $n = 15$ for 15 T + 1B). White arrows show the sites of budding events. All data are from three independent experiments. Scale bars are 500 μm in all micrographs. Diagram showing tentacle arrangement is provided under each image. Tentacles and buds are depicted as gray and black discs, respectively. The numbers inside the discs indicate the sequence of budding. **c** Quantification of tentacle number including buds in growing polyps. Tentacle progression is indicated on the top of each graph ($n = 98$, 137, and 85 polyps in three independently growing groups). **d** Quantification of budding sequence in growing polyps. S4s6 > s4s6 means two *trans*-budding events in s4 and s6 segments ($n = 56$, 86, and 40 polyps in three independently growing groups). Data are mean ± SD for error bars (unpaired Student's two-tailed *t*-test, ****$p < 0.0001$, ns nonsignificant, $p = 0.8699$). Source data are provided as a Source data file.

**TOR-dependent growth is required for tentacle formation**. To test the role of growth in nutrient-dependent tentacle budding, we used rapamycin to inhibit the TOR pathway, a growth regulatory module that integrates multiple cellular inputs, including nutrition, energy availability, and growth factor signaling[36]. Polyps were fed daily for 8 days, while concomitantly treated with 1 μM rapamycin. As expected, control animals exhibited increased body size and developed tentacle buds (Supplementary Fig. 5). In contrast, animals treated with rapamycin did not grow and failed to form new buds, although they internalized the orange pigment of *Artemia* as an indicator of successful feeding (Supplementary Fig. 5). Consistent with a growth defect, cell proliferation was dramatically reduced in rapamycin-treated animals (Fig. 5c, d and Supplementary Fig. 8). Lack of bud formation was also observed when polyps were only treated with rapamycin from day 3,

during the expected period of tentacle budding (Supplementary Fig. 8). These results show a continuing requirement for TOR-dependent growth in postembryonic tentacle development.

To visualize the activity of the TOR pathway, we utilized an antibody directed against a conserved phosphorylation motif in the 40S ribosome protein S6 (pRPS6, Supplementary Fig. 9)[37]. This protein is a direct substrate of ribosomal protein S6 kinase (S6K), and its phosphorylated form is a reliable marker for TOR complex 1 and S6K activation[38]. In unfed polyps, cytoplasmic pRPS6 staining was not detected in either epidermal or gastrodermal layers, but pRPS6 immunoreactivity was mainly localized at the apical regions of epithelial cells (Fig. 5a). Following feeding, we observed ubiquitous cytoplasmic pRPS6 staining in the epidermal layer that reflected the active metabolic state of growing polyps (Fig. 5b and Supplementary

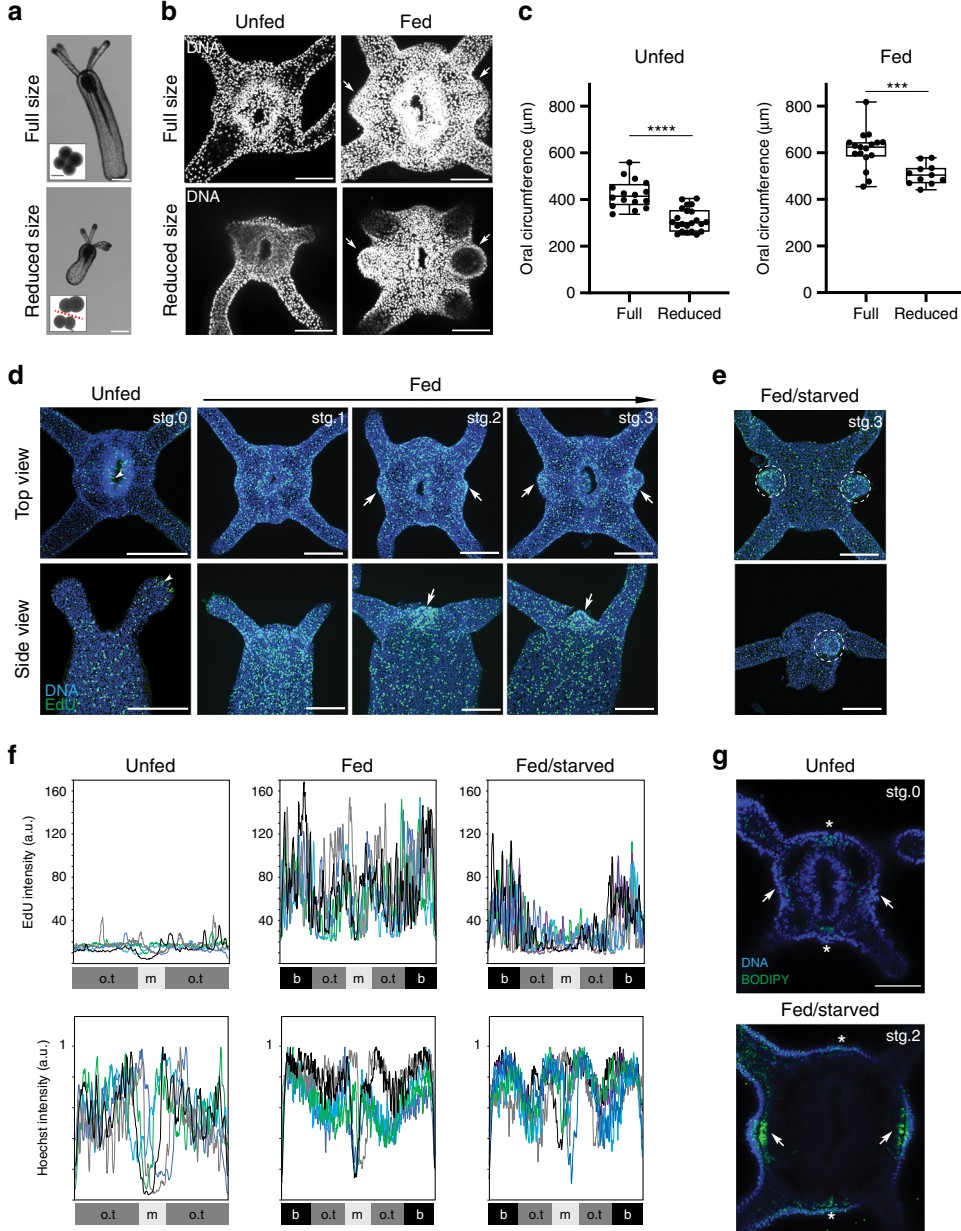

**Fig. 4 Global and polarized growth precede tentacle addition. a** Upper panel: unfed full-sized primary polyp. Inset box shows a four-cell stage embryo. Lower panel: unfed reduced-sized polyp resulting from blastomere isolation. Inset box shows four-cell stage embryo divided into two pairs of blastomeres (two independent experiments). Scale bars are 100 μm. **b** Confocal z-projection of top views for the oral poles stained with Hoechst (white) in the indicated conditions. White arrows show tentacle buds. Scale bars are 50 μm. **c** Quantification of oral circumferences (unfed n = 16 and fed n = 17 full-sized polyps; unfed n = 22 and fed n = 11 reduced-sized polyps). Data are shown as individual data points for each polyp in a Box and Whiskers graph (bottom: 25%; top: 75%; line: median; whiskers: min to max; unpaired Student's two-tailed t-test, ****p < 0.0001, ***p < 0.001). **d, e** Confocal projections of animals stained for EdU incorporation (green) and with Hoechst (blue) in two independent experiments. White arrows indicate localized enrichment of EdU incorporation. White arrowheads indicate examples of unspecific EdU-labeling at the tentacle tip and in the pharynx. Dashed circles show tentacle buds maintaining EdU, while the animals were starved. Fed/starved means animals were fed for 3 days than starved for 4 days. Budding stages from 0 to 3 are indicated (see Supplementary Fig. 6). Scale bars are 100 μm. **f** Plots of EdU intensity and normalized Hoechst intensity across the oral tissue (o.t.) of segments s3/s7, mouth (m), and bud (b) areas. Each colored line represents quantification in a single animal (unfed polyps n = 5, fed polyps n = 5, and fed/started polyps n = 6, a.u. arbitrary units). Note that fed animals show the highest amplitude of EdU intensity, which probably reflects an increased rate of Edu incorporation during S-phase due to a fast cell cycle. **g** Confocal z-stacks of animals stained with BODIPY (green) and Hoechst (blue; unfed: n = 6 polyps and fed/starved n = 8 polyps in two independent experiments). White arrows and asterisk indicate the sites of the first and second trans-budding, respectively. Scale bar is 50 μm. Source data are provided as a Source data file.

Fig. 8). Interestingly, stronger cytoplasmic pRPS6 staining marked bud primordia in both germ layers and provided a clear molecular readout of tentacle patterning (Fig. 5b and Supplementary Fig. 8). Confirming the TOR dependence of RPS6

phosphorylation, fed animals exposed to rapamycin exhibited a dramatic reduction of pRPS6 and failed to phosphorylate RPS6 at presumptive tentacle primordia (Fig. 5c, e and Supplementary Fig. 8). Taken together, these results show that feeding induces

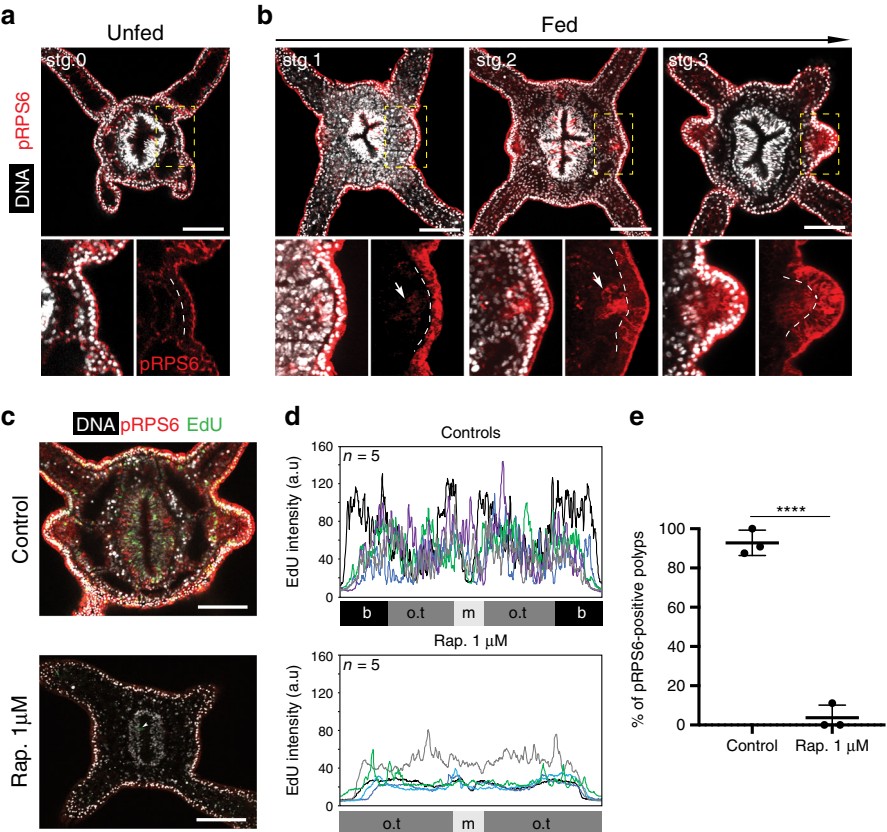

**Fig. 5 TOR pathway is required for growth and tentacle budding in fed polyps. a** Confocal *z*-section of the oral pole of unfed polyp stained with with an antibody against pRPS6 (red) and Hoechst to label nuclei (white) in four independent experiments (*n* = 28 polyps). Inset box is a zoom in of the segment 3. **b** Confocal sections of developing buds at sequential stages (*n* = 32 polyps). Inset boxes show a higher magnification of segment 3 for each stage. Dashed lines separate the outer- and inner-layer nuclei. Arrows indicate the enrichment of pRPS6 in the inner-layer of tentacle primordia. **c** Confocal projection of the oral pole of 5 days fed control and 8 days fed rapamycin-treated polyps stained for EdU incorporation (green), an antibody against pRPS6 (red) and with Hoechst (white; three independent experiments). White arrowhead indicates an unspecific EdU-labeling in the pharynx. **d** Plots of EdU intensity in control polyps (*n* = 5) and rapamycin-treated polyps (*n* = 5). Scale bars are 50 μm. **e** Quantification of polyps showing PS6RP-postive tentacle primordia in control polyps (*n* = 25) and rapamycin-treated polyps (*n* = 26). Data are mean ± SD for error bars (unpaired Student's two-tailed *t*-test, ****p < 0.0001). Source data are provided as a Source data file.

the organismal activation of the TOR pathway, which becomes spatially patterned and defines the location of tentacle primordia.

**Fgfrb expression marks tentacle primordia in polyps**. Based on the results described above, we hypothesized that TOR-dependent organismal growth modulates the activity of developmental signaling pathways, which in turn generates a feedback loop that locally enhances TOR pathway activity and promotes polarized growth in tentacle primordia. FGFR signaling is an attractive candidate to mediate this function, as discrete cell clusters expressing *Fgfrb* prefigure the position of tentacle primordia in unfed polyps (Fig. 6a and Supplementary Movie 1)[39]. To define the identity of these orally scattered cells, we generated a reporter line expressing eGFP under the control of ~8.7 kb surrounding the promoter region of *Fgfrb* (Supplementary Fig. 10 and Supplementary Movie 2). By combining fluorescent in situ hybridization of *Fgfrb mRNA* and α-eGFP immunostaining, we confirmed the overlap between the endogenous and transgenic *Fgfrb* expression in primary polyps (Fig. 6c and Supplementary Fig. 10). Based on the morphology of eGFP-positive cells and F-actin staining, we assigned these *Fgfrb*-positive cell clusters to a subpopulation of ring muscle cells, characterized by an epitheliomuscular architecture that display elongated basal processes associated with myofilaments (Fig. 6c and Supplementary Fig. 10)[40]. Interestingly, these *Fgfrb*-positive cell clusters were also found during larval development,

indicating their potential pre-metamorphic origin (Supplementary Fig. 10). At the oral pole, the *Fgfrb-eGFP* line also labeled longitudinal muscles and pharyngeal cells, which is consistent with the endogenous expression pattern of *Fgfrb* (Supplementary Fig. 10).

To determine the effect of feeding on *Fgfrb* expression, we performed fluorescent in situ hybridization in growing and budding polyps (Fig. 6a, b). In fed polyps, the expression of *Fgfrb* expanded from a small number of cells in the inner layer to a larger domain within the body segment (Fig. 6a, b). This feeding-dependent expansion was nucleated around the initial *Fgfrb*-positive ring muscles, as visualized by the different temporal dynamics between *Fgfrb* mRNA and *Fgfrb-eGFP* expression (Fig. 6d and Supplementary Fig. 10). Interestingly, pRPS6-positive cells also adopted a similar organization surrounding the pre-feeding *Fgfrb*-positive cells (Fig. 6e and Supplementary Fig. 10). During budding, both epidermal and gastrodermal cell layers showed an enrichment of *Fgfrb* expression that overlapped with pRPS6 staining in tentacle primordia (Fig. 6b, e). Similar to TOR pathway activity, *Fgfrb* expression showed a segment-dependent regulation in response to feeding (Supplementary Fig. 10). Segments s3 and s7 were first to express this pattern, which mirrored the sequence of the first *trans*-budding event. Fed polyps treated with 1 μM rapamycin did not show an expansion of *Fgfrb* expression, while they maintained the discrete cell clusters expressing *Fgfrb* (Fig. 6f). Taken together, these results

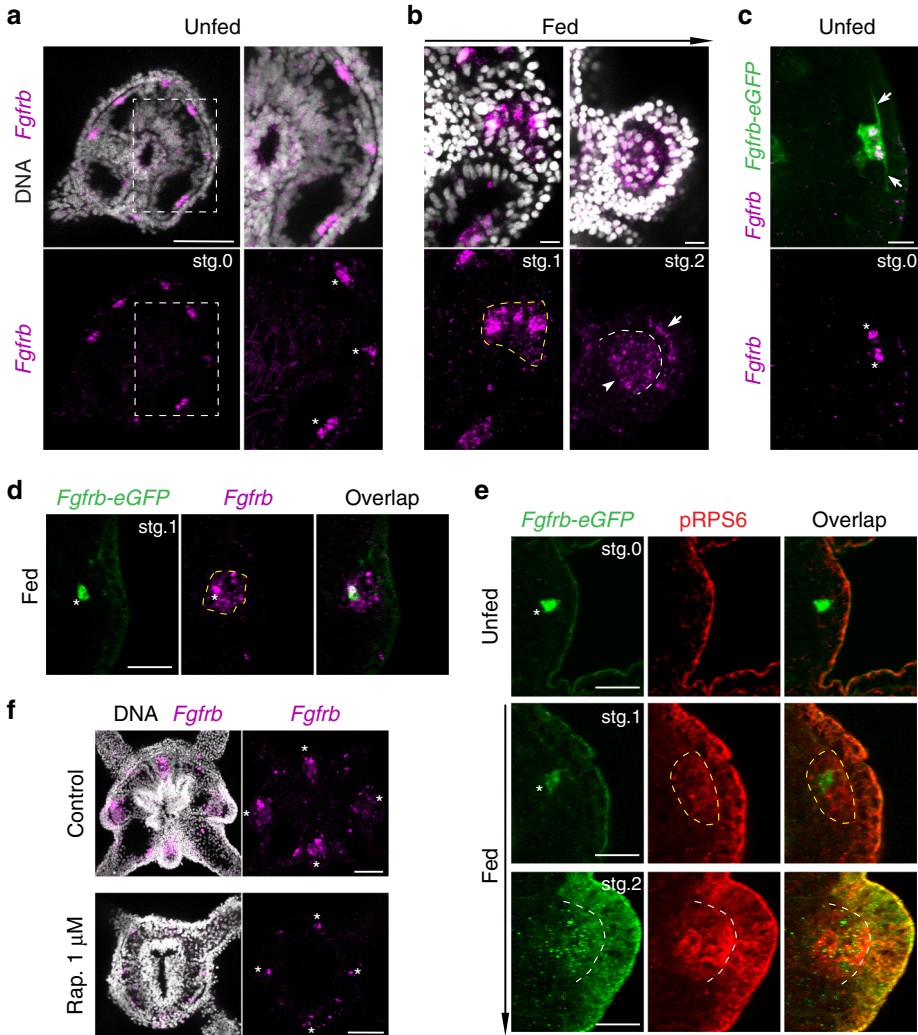

**Fig. 6 *Fgfrb*-positive ring muscles pre-mark the sites of tentacle primordia. a, b** Fluorescent in situ hybridization of *Fgfrb* (unfed polyps $n = 8$, fed polyps $n = 14$, two independent experiments). **a** Confocal *z*-projections of the oral pole of unfed polyp showing *Fgfrb* expression (purple) and nuclei (white). Inset box is a zoom in of s3 segment. **b** Confocal *z*-projections of developing buds at sequential stages. White arrow and arrowhead indicate outer- and inner-body layer, respectively. **c, d** Confocal *z*-projections of the *Fgfrb-eGFP* transgenic line stained with α-eGFP (green) and labeled for *Fgfrb* mRNA (purple) (unfed polyps $n = 6$, fed polyps $n = 5$, two independent experiments). White arrows show the elongated basal parts of ring muscles. White arrowheads indicate nuclear localization of *Fgfrb* mRNA in the eGFP-positive cells. Yellow dashed line delineates the expression domain of *Fgfrb* after feeding. **e** Confocal *z*-projections of unfed and fed *Fgfrb-eGFP* polyps stained with α-eGFP (green) and pRPS6 antibody (red) in three independent experiments. Yellow dashed line delineates the domain of pRPS6-positive cells. White dashed line separates the two body layers. **f** Confocal projection of the oral pole of fed controls and fed rapamycin-treated polyps stained with Hoechst (white) and labeled for *Fgfrb* mRNA (purple). Note the expansion of *Fgfrb* expression in tentacle primordia in controls ($n = 8$ polyps), while there is no change in *Fgfrb* expression in rapamycin-treated polyps ($n = 11$ polyps) in two independent experiments. White asterisks indicate *Fgfrb*-expressing cells in the oral segments. Scale bars are 100 μm. Scale bars are: 50 μm in **a** and **f**, 10 μm in **b** and **c**, and 20 μm in **d** and **e**.

show that there are at least two early events associated with the feeding-dependent tentacle development. First, ring muscle cells expressing *Fgfrb* pre-mark the sites of the morphogenetic changes in tentacle primordia. Second, TOR activity drives the feeding-dependent expansion of *Fgfrb* in neighboring cells.

***Fgfrb* controls axial elongation and feeding-induced budding.**
To determine the role of *FGFR* signaling, we first treated fed polyps with the FGFR signaling inhibitor SU5402 (Supplementary Fig. 11)[41]. While control and drug-treated polyps showed a similar size increase following 5 days of development, the expected spatial enrichment of pRPS6 staining and cell proliferation at tentacle primordia was not detected in

SU5402-treated polyps (Supplementary Fig. 11). As SU5402 can inhibit both PDGF and VEGF signaling[42], we next genetically disrupted *FGFRb* signaling. To do so, we generated independent mutant alleles of *Fgfrb* using the CRISPR/Cas9 system. Several alleles were isolated, including two putative null alleles *Fgfrb^{mut1}* and *Fgfrb^{mut2}*, which disrupted the first and second coding exons, respectively (Fig. 7a). To minimize any possible CRISPR/Cas9 off-target effects that could be expressed in homozygous animals, we analyzed the phenotypes of F2 *trans*-heterozygous *Fgfrb^{mut1}*/ *Fgfrb^{mut2}* individuals. Crossing F1 heterozygotes resulted in Mendelian ratios of viable F2 *trans*-heterozygous polyps, whose genotypes were confirmed by DNA sequencing (Supplementary Fig. 12). *Fgfrb^{mut1}*/*Fgfrb^{mut2}* animals exhibited a significantly reduced length of both body column and tentacles compared to

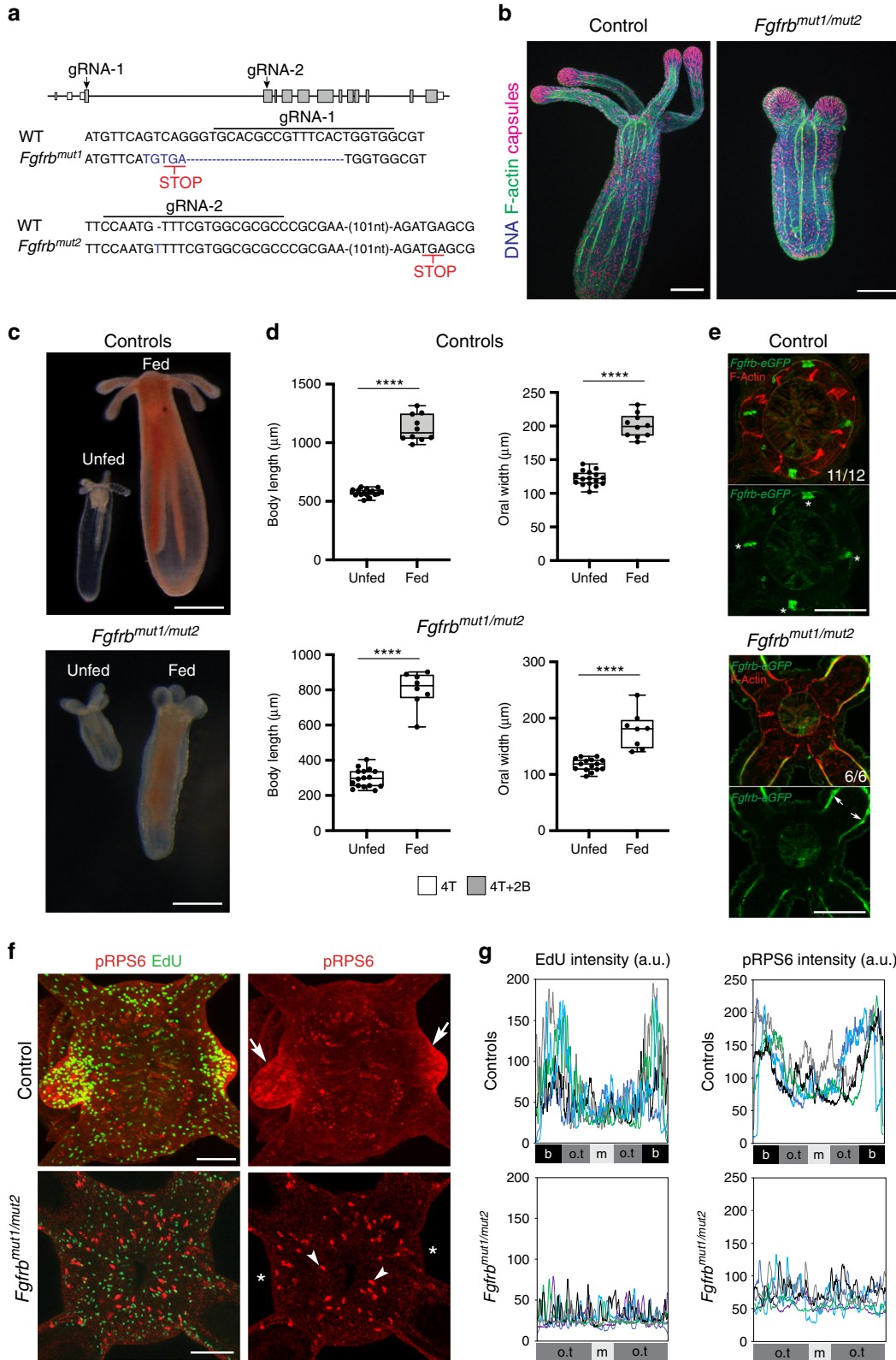

sibling controls (Fig. 7b and Supplementary Fig. 12). This phenotype most likely resulted from a failure of *Fgfrb* mutants to properly elongate during metamorphosis. In addition, *Fgfrb* mutants also displayed reduced septal filaments and defects in longitudinal tentacle muscle compared to their siblings (Supplementary Fig. 12). Nevertheless, these mutants exhibited the expected eight segments and four primary tentacles with

cnidocyte enrichment at the tip, indicating normal proximo-distal patterning (Fig. 7b, e).

As *Fgfrb* mutants were viable, we tested their ability to resume development in response to feeding. Out of 20 hand-fed mutants, eight polyps consistently consumed food and exhibited growth, although a tripling of their initial body length took much longer (~6 weeks versus 2 weeks in controls, Fig. 7c, d). This delayed

**Fig. 7 *Fgfrb* signaling is required for the feeding-dependent tentacle development. a** Gene model of the *Fgfrb* locus showing the target sites of gRNA1 and gRNA2, and resulting indel mutations that create premature stop codons (red). PAM motifs are in bold characters. **b** Confocal *z*-projections of indicated polyps stained with phalloidin (F-actin, green) and DAPI (nuclei, blue) in two independent experiments. DAPI also stains mature nematocyst capsules (purple). Scale bars are 100 µm. **c** Image of unfed and fed indicated polyps. Scale bars are 250 µm. **d** Quantification of body length and oral width in control (unfed, $n = 15$ polyps; fed, $n = 10$ polyps) and *Fgfrb* mutant polyps (unfed, $n = 15$ polyps; fed, $n = 8$ polyps). Animals were imaged live in two independent experiments. The number of tentacles and buds is indicated. Data are shown as individual data points for each polyp in a Box and Whiskers graph (bottom: 25%; top: 75%; line: median; whiskers: min to max; Unpaired Student's two-tailed *t*-test, ****$p < 0.0001$). **e** Confocal *z*-projections of oral poles of indicated polyps stained with α-eGFP (green) and phalloidin (red). White asterisks indicate ring muscles expressing *Fgfrb*. Note that these discrete ring muscles cells, but not *Fgfrb*-positive longitudinal tentacle muscles, are missing in the mutant background (controls $n = 12$ polyps and *Fgfrb* mutants $n = 6$ polyps).
**f** Confocal *z*-projections of oral poles of indicated polyps stained for EdU incorporation (green), an antibody against pRPS6 (red), and with Hoechst (blue) in two independent experiments. Scale bars are 50 µm. White arrows indicate the first *trans*-budding event marked by pRPS6 in fed polyps. White asterisks (*) show the expected sites for tentacle bud formation. White arrowheads indicate isolated puncta of pRPS6 staining in the *Fgfrb* mutant. These unknown puncta are less prominent in control animals. **g** Quantification of EdU and pRPS6 intensity in controls ($n = 5$ polyps) and mutants ($n = 6$ polyps); a.u. arbitrary units. In all panels, control animals are siblings of the *Fgfrb* mutants. Source data are provided as a Source data file.

growth was not observed in SU5402-treated polyps (Supplementary Fig. 11), suggesting that this phenotype might be the result of disrupting *Fgfrb* function during embryonic–larval development. Still, despite the significant organismal growth of *Fgfrb* mutants, they failed to elongate the four primary tentacles and did not develop new tentacle buds (Fig. 7c, d). Further, these mutants showed shorter and wider primary tentacles compared to controls. Consistent with the *Fgfrb*-dependent patterning defect in postembryonic tentacle development, mutants lacked the oral *Fgfrb-positive* cell clusters as assessed with both endogenous mRNA and transgenic *Fgfrb* expression (Fig. 7e, Supplementary Fig. 12, and Supplementary Movies 3 and 4). In addition, *Fgfrb* mutants did not exhibit pRPS6-positive domains or polarized growth in the expected segments (Fig. 7f, g). Taken together, we conclude that *Fgfrb* signaling is dispensable for primary tentacle budding, but is essential to pattern TOR pathway activity and localized cell proliferation in subsequent tentacle primordia.

## Discussion

A diversity of organisms has evolved strategies to both sense and adapt to changes in their environment, resulting in the remarkable plasticity of development. Among animals, environmental plasticity has been widely described in ephemeral species (e.g., insects, worms, and vertebrates). These animals typically experience specific periods, in which development is responsive to environmental cues, producing long-lasting changes in either form or physiology[43]. In contrast, species with extreme longevity must continuously adjust their developmental behaviors to unpredictable fluctuations of food supply. This plasticity underlies diverse adaptive phenomena, such as reversible body size changes in adult planarians[44], or asexual reproduction in *Hydra*[45]. Along similar lines, here we uncovered the cellular and signaling mechanisms by which *Nematostella* polyps integrate the nutritional status of the environment to control postembryonic tentacle development.

Our results establish the spatial map for tentacle addition in the sea anemone *Nematostella*. By examining a large number of polyps progressing from the 4- to the 16-tentacle stage, we showed that two budding modalities, *cis* and *trans*, drive tentacle addition in a stereotyped spatial pattern. While the mechanisms that direct the temporal sequence of budding are unknown, the initial step of this process is the formation of outgrowths from radial segments that lack primary tentacles. Once each segment developed a single tentacle, secondary tentacle territories sequentially emerged within select segments, except in the directive segments s1 and s5. These territories preceded tentacle development and were defined by the formation of short gastrodermal folds enriched in F-actin. Based on these observations and as previously reported during Hox-dependent embryonic

segmentation[20], the formation of endodermal territories is a common theme in tentacle patterning. However, whether Hox genes play a central role in subdividing preexisting segments in polyps remains unknown.

TOR signaling is a major regulator of growth and RPS6 is an evolutionary conserved target of this pathway in eukaryotes[36,38]. Our findings show that tentacle development in polyps is preceded by a feeding-dependent global growth phase, followed by the formation of localized cell proliferation zones that define tentacle bud sites. Both patterns of growth correlate with the phosphorylation of RPS6, which is highly enriched in developing tentacles compared to the rest of the body. However, our results also suggest that distinct upstream inputs co-regulate cell proliferation and the phosphorylation of RPS6. These findings are consistent with the distinct sensitivity of these two processes to starvation. While the uniform pattern of cell proliferation is primarily dependent on feeding, the localized cell proliferation and phosphorylation of RPS6 at polyp tentacle primordia selectively requires *Fgfrb* function. We propose that *Nematostella* translates nutrient inputs into organismal growth, which in turn modulates the pre-defined developmental signaling landscape of body segments and promotes postembryonic tentacle development. Interestingly, discrete ring muscle cells expressing *Fgfrb* pre-mark the sites of postembryonic primordia and nucleate the early morphogenetic events associated with the feeding-dependent tentacle development. Consistent with the pattern of tentacle addition, the *Fgfrb*-positive ring muscles do not simultaneously engage in postembryonic tentacle development, suggesting that there is an unknown mechanism that controls their deployment in time and space. In the flatworm, muscle fibers can encode positional information that is critical for guiding tissue growth and regeneration[46,47]. While this property of muscles has not been reported in cnidarians, the feeding-dependent tentacle development in *Nematostella* offers the opportunity to explore the developmental function of ring muscles.

In contrast to the tentacle development in polyps, cell proliferation is not spatially patterned during embryonic tentacle morphogenesis[4]. This difference suggests that there are distinct morphogenetic trajectories leading to embryonic and nutrient-dependent postembryonic tentacle development. The *Nematostella* genome contains two FGF receptors (a and b) and 15 putative FGF ligands[39,41]. The function of the *Nematostella* FGF signaling pathway has only been investigated during embryonic and larval development[41]. Based on knockdown experiments, *Fgfra* is essential for apical organ formation and metamorphosis. In the current study, we generated a stable mutant line for *Fgfrb*. Phenotypic analysis of *Fgfrb* shows that mutant larvae undergo metamorphosis, but the process of axial elongation is disrupted. Interestingly, FGF signaling plays a critical role in the elongation

of vertebrate embryos[48]. While further investigation is required to define the specific role of *Fgfrb* during embryonic *Nematostella* development, this finding highlights a potential pre-bilaterian function of FGFR signaling in body elongation. These results also reveal that *Nematostella* FGF receptors have distinct functions during development, suggesting the subfunctionalization of paralogs, with *Fgfra* having a dominant role during embryonic development. On the other hand, *Fgfrb* mutant polyps are viable and exhibit the four primary tentacles. With careful feeding and attention, these mutants can grow to some extent, but they do not show the localized cell proliferation and phosphorylation of RPS6 that are characteristic of polyp tentacle primordia. In sum, this finding reveals that FGF signaling couples the nutrient-dependent organismal growth with postembryonic tentacle development.

## Methods

**Animal husbandry and stereomicroscopic imaging.** *N. vectensis* were cultured in 1/3 artificial seawater (Sea Salt; Instant Ocean). Adult polyps were spawned using an established protocol[15]. Progeny from nine spawning events were segregated into three groups, each containing ~1500 animals. Those groups were then further subdivided into more manageable populations of 250–350 animals. Each dish was fed *Artemia* nauplii two to three times per week. To ensure culture quality, seawater and dishes were changed weekly and biweekly, respectively. Tentacle pattern was monitored at different post-feeding time points, and polyps were selected for fixation when new tentacle stages were observed. Selected polyps were relaxed in 7% MgCl$_2$ (Sigma-Aldrich) and fixed in 4% paraformaldehyde (PFA; Electron Microscopy Sciences) for 90 min at room temperature. Larger animals were fixed in 4% PFA overnight at 4 °C. The fixed animals were washed with PBS and stored at 4 °C until imaged. To orient the oral pole of fixed polyps along the directive axis, two morphological features were used. The position of primary mesenteries served as a landmark to orient polyps progressing from 4 to 12 tentacles. However, as the animals grow, this morphological feature became less pronounced in adults bearing >12 tentacles. At these stages, the location of siphonoglyph was more visible and was used to define the polarity of the directive axis. For imaging, polyps were decapitated and directly imaged in the Petri dish using a Leica MZ16F stereo-microscope with a QImaging QICAM *FAST* 12-bit color camera.

**Feeding of primary polyps and drug treatment.** Prior to feeding, developing animals were raised for 3–4 weeks at 23 °C in 1/3 artificial seawater in the dark. To perform feeding, ~300 μl of concentrated *Artemia* was partially homogenized and mixed with 40–50 primary polyps in a 6-cm Petri dish. One day of feeding corresponded to the incubation of polyps with *Artemia* for ~3 h followed by a water change. For drug treatments, rapamycin (1 μM; Sigma, R8781) and SU5402 (20 μM; Sigma, SML0443) were applied in 0.2% DMSO in artificial seawater at room temperature in the dark and were refreshed daily post-feeding. Concurrently, control animals were incubated in 0.2% DMSO in artificial seawater.

**Immunohistochemistry and staining.** Polyps were incubated with EdU (300 μM from a stock dissolved in DMSO) in artificial seawater for 30 min at room temperature (Click-it Alexa Fluor 488 Kit; Molecular Probes)[49]. After incorporation, animals were relaxed in 7% MgCl$_2$ in artificial 1/3 seawater for 10 min, fixed in cold 4% PFA (Electron Microscopy Sciences) in 1/3 artificial seawater for 1 h at room temperature. Samples were washed three times in PBS and permeabilized in PBT (PBS with 0.5% Triton X-100; Sigma) for 20 min. The reaction cocktail was prepared based on the Click-it Kit protocol and incubated with the animals for 30 min. After three washes in PBS, samples were labeled with Hoechst 34580 (1 μg ml$^{-1}$; Molecular Probes) in PBT overnight at 4 °C. When combined with immunostaining, EdU-labeled polyps were stained with primary (rabbit anti-pRPS6 Ser235/236, 1:50; Cell Signaling #4858) and secondary (goat anti-rabbit IgG Alexa Fluor 594, 1:500; Molecular Probes) antibodies in the blocking buffer (PBS with 10% goat serum, 1% DMSO, and 0.1% Triton X-100) at 4 °C overnight[4]. For phalloidin and DAPI staining, polyps were fixed in 4% PFA with 10 mM EDTA for 1 h and staining was carried out overnight followed by four washes with PBT[4].

For BODIPY staining, fixed animals were incubated with BODIPY® 493/503 (1 μg ml$^{-1}$; ThermoFisher Scientific) for 60 min and then washed four times with PBS. To image the oral view of polyps, specimens were incubated in 87% glycerol (Sigma), decapitated with a sharp tungsten needle and mounted on glass slides with spacers. All images were taken with Leica SP5 or SP8 confocal microscopes. Oil Red O staining was performed using an established protocol[50]. Unfed and fed primary polyps were relaxed in 7% MgCl$_2$ for 5 min and fixed for 1 h in 4% PFA in 1/3 seawater at room temperature. Samples were then washed twice in PBS for 5 min followed by 30 s in 60% isopropanol. Oil Red O staining was performed for 30 min at room temperature using a fresh and prefiltered solution of 3 mg ml$^{-1}$ Oil Red O (Sigma) in 60% isopropanol. The staining was stopped by adding 60% isopropanol and two washes in PBS for 5 min. Polyps were incubated in 80% glycerol and imaged using LEICA MZ 16 F microscope.

**RNA in situ hybridization.** The RNA in situ probe for *Fgfrb* was designed to cover ~900 nucleotides. cDNA of *Fgfrb* was cloned from total RNA isolated from mixed stages of animals using the RNeasy Mini Kit (Qiagen). Amplification primers were: *Fgfrb*_fwd 5′-AAACGCGAAAAGACCCTGATAGC-3′ and *Fgfrb*_rev 5′-GGA-CAGCGGGGACGTCAG-3′ antisense probe was synthesized by in vitro transcription (MEGAScript Kit; Ambion) driven by T7 RNA polymerase with DIG incorporation (Roche). Chromogenic and fluorescent RNA in situ hybridization were carried out using established protocols[51,52]. Primary polyps were relaxed in 7% MgCl$_2$ solution for 5 min before fixation in 0.25% glutaraldehyde/4% PFA/seawater for 1–2 min at 4 °C, followed by 4% PFA for 1 h at 4 °C. Fixation was followed by five washes of 5 min in PTW (PBS with 0.1% Tween) and dehydration in an ethanol series. After rehydration in 60 and 30% ethanol, and five washes in PTW, samples were digested by 10 μg ml$^{-1}$ proteinase K in PTW for 20 min and stopped by washing twice in glycine (4 mg ml$^{-1}$ in PTW). Samples were then treated with 0.1 M TEA (2×, 5 min), 0.1 M TEA/0.3% (v/v) acetic anhydride (5 min), 0.1 M TEA/0.6% acetic anhydride (5 min), PTW (2×, 5 min) followed by re-fixation in 4% PFA/PTW for 1 h. Fixation was stopped by five washes in PTW and 10 min in 50% hybridization solution (HS)/PTW before adding HS (50% (v/v) formamide, 1% (w/v) SDS, 5× SSC (pH = 4.5), 0.05 mg ml$^{-1}$ heparin, 0.1 mg ml$^{-1}$ salmon testes DNA, and 0.1% (v/v) Tween in DEPC-treated water). After a 10 min incubation, HS was refreshed and samples were incubated at 60 °C for at least 2 h. Hybridization took place for 16–20 h at 60 °C in HS containing DIG-labeled RNA probe (0.5–1.5 ng μl$^{-1}$). Post-hybridization washes at 60 °C were as follows: 10 and 40 min in 100% HSW (HSW is HS without salmon testes DNA), 30 min each in 75%/25%, 50%/50%, and 25%/75% HSW/2× SSC, 30 min in 2× SSC and 3× 20 min in 0.2× SSC, followed by 10 min washes at room temperature in 75%/25%, 50%/50%, and 25%/75% 0.2× SSC/PTW and 100% PTW.

For colorimetric detection, samples were washed five times in PBT (0.1% BSA + 0.2% triton in PBS) and blocked for 1 h at 4 °C with blocking buffer (1% Roche Blocking Reagent in MAB) before overnight incubation at 4 °C with 1:4000 αDIG-AP (Roche) in blocking buffer. Ten washes of 20 min each in PBT and three washes of 10 min in NTMT(L) (50 mM MgCl$_2$, 100 mM NaCl, 100 mM Tris-HCl pH = 9.5, 0.1% Tween), were followed by staining in NTMT(L) containing NBT/PCIP (Promega). Background color was removed by two washes in PTW followed by overnight incubation in ethanol and five washes of 5 min in PTW before mounting in 80% glycerol.

For fluorescent detection, endogenous peroxidases were blocked for 1 h in 3% H$_2$O$_2$ and animals washed five times for 5 min each in PBT before blocking for 1 h at 4 °C, with blocking buffer and o/n incubation at 4 °C in 1:1000 αDIG-POD (Roche) in blocking buffer. Ten washes of 20 min each in TNT (0.1 M Tris-HCl, 0.15 M NaCl, and 0.05% Tween) were followed by a 40 min detection step using the TSA Plus-Cy5 Fluorescence Kit (Perkin Elmer). Samples were washed four times 10 min each in TNT before staining of cell nuclei with Hoechst and mounting in 80% glycerol. Following fluorescent in situ hybridization, immunostaining was performed as described above[53]. Bright-field images were acquired under a Leica DM4000 microscope and confocal images were taken with a Leica SP8 microscope.

**CRISPR/Cas9 mutagenesis and transgenesis.** CRISPR/Cas9 genome editing in *Nematostella* embryos was carried out using an established protocol[25]. Guide RNAs (gRNAs) were designed using the online web interface http://chopchop.cbu.uib.no. Two gRNAs targeting the first and second coding exons were generated via PCR reaction and purified using QIAquick PCR Purification Kit (QIAGEN). The sequence targets were: first coding exon: GGTGCACGCCGTTTCACTGG**TGG** and second coding exon: **CCA**ATGTTTCGTGGCGCGCCCGC. gRNAs were in vitro transcribed using the MEGAshortscript T7 kit (Life Technologies) and purified using 3 M sodium acetate/ethanol precipitation. Recombinant Cas9 protein with NLS sequence (800 ng μl$^{-1}$; PNA Bio, #CP01-20) was co-injected with each gRNA (500 ng μl$^{-1}$) into unfertilized *Nematostella* oocytes. Injected oocytes were then fertilized and raised at room temperature.

To test the efficiency of genome editing in F0 injected animals, genomic DNA was extracted from individual primary polyps[25]. Each polyp was then incubated in 20 μl DNA extraction buffer (10 mM Tris-HCl pH 8, 50 mM KCl, 0.3% Tween 20, 0.3% NP40, and 1 mM EDTA) containing 1 μg μl$^{-1}$ proteinase K at 58 °C for at least 5 h. Proteinase K was heat-inactivated by incubating the reaction at 98 °C for 20 min. We used 2 μl of genomic DNA extract for subsequent PCR analysis. Following sequencing confirmation of indel or deletion events, F0 injected animals with tentacle defects were raised to sexual maturity and crossed to wild-type animals. The progeny of these crosses was raised and individually genotyped using genomic DNA extracted from approximately five surgically isolated tentacles. Sequenced PCR products showing overlapping peaks in their chromatograms were cloned using NEB PCR cloning kit (NEB #E1202S) and then sequenced to define the identity of mutations carried by F1 heterozygous animals. For each gRNA-induced lesion, F1 animals carrying identical frameshift mutations were grouped by sex to form spawning groups. To confirm the relationship between genotype and phenotype, F2 progeny resulting from the crosses of F1 heterozygous animals were individually genotyped as described above.

The *Fgfrb-eGFP* transgenic line was generated by meganuclease-mediated transgenesis[26]. The genomic region surrounding the *Fgfrb* promoter (coordinates: Scaffold 4; 2003298–2012061) was cloned in the transgenesis plasmid[26] and the first coding exon of *Fgfrb* was replaced by eGFP with the SV40 poly A, using

NEBuilder HiFi DNA assembly (NEB, E2621). F2 *Fgfrb-eGFP* homozygotes were established and crossed to *Fgfrb* heterozygous animals to combine the mutant alleles with the *Fgfrb* reporter construct. In all experiments, the expression of eGFP was visualized using immunostaining with anti-eGFP antibody (mouse, ThermoFisher, A-11120).

**Quantification of EdU intensity and body dimensions**. Images were analyzed with ImageJ software[54]. EdU and Hoechst signals were quantified by the total intensity in a selected area of the oral pole spanning s3/s7 segments. The oral area was delineated with the straight line tool with 25 μm width. For body dimensions, body length was measured from the oral opening to the aboral end. Body width was measured at the base of the tentacles. A straight line tool was used for both measurements. Oral circumference was defined with the oval selection in z-projection images of oral views.

**Reporting summary**. Further information on research design is available in the Nature Research Reporting Summary linked to this article.

## Data availability

We declare that the main data supporting the findings of this study are available within the article and its Supplementary Information files. Source data are available in the Source data file for Figs. 2, 3, 4, 5 and 7, and Supplementary Figs. 3, 5, 11 and 12. Extra data are available from the corresponding author upon request. Source data are provided with this paper.

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

## Acknowledgements

We thank the Stowers institute and EMBL aquatic cores for animal husbandry, as well as ALMF at EMBL for imaging support. We also thank A. Ephrussi, A. Aulehla, and T. Hiiragi for discussion and comments on the manuscript. This work was supported by the Stowers Institute for Medical Research and European Molecular Biology Laboratory. M.A. is supported by Human Frontier Science Program LTF (LT00126/2019-L).

## Author contributions

A.I designed and conceptualized the study, and wrote the manuscript. M.C.G. also designed the study and edited the manuscript. A.I generated the mutant and reporter lines. M.R.M. and A.I. built the spatio-temporal map of tentacle addition. P.J.S. performed in situ hybridization experiments, genotyping, and genetic crosses. P.J.S. and M.A. performed all drug treatment and staining experiments. L.R.E. raised founders and heterozygous mutant animals to sexual maturity. A.S. performed image processing.

## Funding

## Competing interests

The authors declare no competing interests.
