## [Peer Review File · Nature Communications]

Reviewers' Comments:

Reviewer #1:

Remarks to the Author:

Ikmi et al. describe an intriguing feature of postembryonic development in *Nematostella vectensis*, the addition of tentacles during the growth phase of the animals. They describe the pattern in which pairs of tentacles are added after the first four have formed in the transition to the primary polyp. This pattern shows little variation and does not follow a simple sequence of symmetrical additions. The authors then focus on the first pair of new tentacles and show that these require feeding for their formation and that locally elevated levels of phosphorylated RPS6 and an increase in cell proliferation precede their outgrowth. Confirming that pRPS6 reflects a role for TOR signalling, incubation with rapamycin reduces both pRPS6 and EdU labelling at the site of tentacle formation. Lastly, animals mutant for FGFRb fail to form tentacles after the four tentacle stage, despite being able to feed, suggesting that the elevated levels of FGFRb transcript at the sites of tentacle addition are related to a function of FGF signalling in directing tentacle formation in primary polyps.

Overall, this is a very well executed study with beautiful illustrations of interesting and novel observations. Some aspects are probably not surprising, e.g. that growth requires feeding and that this requires TOR signalling. Still, it has to be shown and this has been done convincingly.

Technically, there are only some points that I would like the authors to consider for improvements, all concerning the role of FGFRb.

1. The expression pattern in the colorimetric in situ in Fig S10 A and B (and in the Matus paper) resembles very much a potential in situ artefact of these late planula/early polyp stages. The stainings in S10C and D and in Fig 6 look more convincing, but they seem to be at a later stage. To support the proposed pre-metamorphic origin, would it be possible to test if these spots remain after shRNA injection? It is also not clear to me how the number of these "clusters" relate to the addition of the tentacles. The tentacles appear in pairs, but there seem to be more clusters of cells (also in the movie). This is a point that is not essential for the manuscript, but maybe the "pre-metamorphic" origin of the clusters is not so clear. The authors probably also see a different way to show that this particular staining in the in situ is specific.

2. Using FGFRb mutants for the analysis is great, but maybe the temporal control allowed by SU5402 treatment could be exploited to obtain more information about the temporal requirement of FGF signalling in this context. Obviously, the gene-specificity is lost, but maybe the comparison of the mutants with transient SU5402 treatments would be informative.

3. A cross-section of the FGFRb mutants showing the organization of the gastric cavity and mesenteries would be informative to understand the phenotype better. Also, it would be good to add more information about the development of the phenotype: when does the animals first differ from the sibs?

While these technical questions (and the minor points listed below) do not diminish my excitement for the data, I am more uncertain about the conceptual background and the implications of the manuscript. The authors embed their study in a context of "continuous developmental capacities" (introduction) and axial patterning (abstract). In the discussion, they mention developmental responses to changes in the environment (nutrition). There is a reference to regeneration, but the phenomenon under study is a regular developmental event that comes to an end at the 16 tentacle stage. To me the broader context remained very unclear. To justify publication at this level, the authors would need to make a more convincing point (evolutionary or developmentally) that the data are important beyond the regulation of one step of tentacle addition in this species.

Minor points:

1. On page 3 the authors mention that they here study adult cnidarians, but this is not the case.
2. The short folds are mentioned quite a while before there is a reference to them being shown in

Fig S4. It might be helpful to have this reference earlier. How far do these short folds extend along the primary body axis?

3. Is it unambiguous that Bodipy stains lipid droplets?

4. What is the staining for the capsules?

5. The description of the movie should be clearer. What are the two "episodes" of the movie (before and after rendering?).

Reviewer #2:

Remarks to the Author:

Review on manuscript NCOMMS-19-17704

The authors in this manuscript describe the post-embryonic tentacle patterning in sea anemone *Nematostella*, showing that patterning is dependent on nutrient availability, induction of the mTOR pathway and activation of FGFR developmental signaling pathways.

The first part of the manuscript is a spatial and temporal description of the tentacles formation in *Nematostella* polyps (from 4 primary tentacles to up to 16-tentacles in the adult) and conclude on a stereotyped and reproducible spatial patterning system. The authors then show that development of a full set of tentacles is feeding-dependent, as unfed polyps arrests at the 4-tentacles stage, and also mTOR dependent, by using molecular markers for cell proliferation and TOR activity. Finally they show that the FGFR expression is essential for post-embryonic tentacle formation, by generating CRISPR/Cas9 mutants.

The authors suggest that mTOR-dependent growth activates the FGFR signaling pathway, which in turn generate a feed back loop to locally enhance mTOR pathway. It is not clear from the manuscript that there is a direct activation by mTOR of the FGFR expansion upon feeding. How is the FGFR pattern in presence of rapamycin?

Overall, the data is well presented and convincing, the manuscript is well written. The work is important for the field of patterning in response to environmental cues (here nutrients) as it sets the ground to further explore the molecular mechanisms regulated by mTOR and FGFR.

Minor remarks:

p3: "bauplan" should be in italics

Figure 5F: rapamycin concentration is 3uM and not 1uM as indicated in text?

Figure S2: title should implicitly indicate that these are unfed polyps

Reviewer #3:

Remarks to the Author:

NCOMMS-19-17704

Summary

The authors examine the relationship between feeding, growth, and tentacle morphogenesis in the sea anemone *Nematostella vectensis*. They characterize the timing and sequence of tentacle formation in the juvenile polyp (after development of the four primary tentacles) and show that the onset of juvenile tentacle morphogenesis occurs only after the onset of feeding. Interestingly, they show that the initial process of primary tentacle formation is uncoupled from the process resulting in tentacles 5-16. Further, they suggest a role for the TOR pathway in both general growth and in juvenile tentacle morphogenesis and show that Rapamycin treatment is sufficient to block the formation of juvenile tentacle buds. They propose a role for FGF signaling in the development of specialized fields of cell proliferation and TOR signaling that demarcate the juvenile

tentacle buds and further suggest that FGF is necessary to couple proliferation and TOR signaling during the general process of growth in these animals.

General Concerns

The description of the morphogenesis of tentacles 5-16 is thorough and while these data are novel, the phenomenon described is specific to *Nematostella vectensis* and as such, these results will not likely be of great importance to anyone outside of the *Nematostella* community. Furthermore, the claim that this study characterizes "lifelong axial patterning" is unfounded. In reality, the number of adult tentacles is largely invariant in this species (as noted by the authors and others before them) so the phenomenon they describe amounts to little more than a description of determinate post-embryonic growth. Although they have provided preliminary data to support a role for TOR and FGF signaling in the tentacle budding process, the data they provide are not of sufficient quality (or magnification) to assess the validity of these claims. Finally, throughout the manuscript, the authors use language that is imprecise and misleading and several key references have been excluded. Despite all of this, the amount of work that went into characterizing the spatial and temporal progression of tentacle morphogenesis is not trivial and these observations do constitute a valuable contribution to the continued development of this emerging model system. I hope the authors can provide additional data to support the very interesting potential role of TOR and FgfRb in regulating juvenile tentacle morphogenesis. The data provided with this manuscript are, unfortunately, not of sufficient quality to substantiate their major claims. I have provided specific comments below intended to help improve the clarity and rigor of this study.

Specific Comments

1. "Lifelong axial patterning" is unfounded

It is not clear that tentacle morphogenesis is really feeding-dependent (abstract) or plastic (discussion). Feeding is clearly required to induce the onset of tentacle morphogenesis but the authors have not demonstrated that animals will arrest at a number of tentacles below 16 if they are starved. Indeed, this study shows that starvation reduces the number of proliferating cells throughout the body, but not in the tentacle bud primordia (Figure 4), which explicitly argues against the claim that this process is feeding-dependent. Also, the term plasticity does not apply when the total number of adult tentacles is determined genetically; unless the authors can demonstrate that tentacles are lost when nutrients are limited or that nutrient limitation in the juvenile stage translates into a permanent change in adult tentacle number, then the use of the term plasticity is inappropriate. Similarly, the authors suggest (page 7) that tentacle budding "scales with body size" despite showing that budding is a binary response to feeding, independent of body size. As presented, there is no validity to this claim either.

2. Imprecise language

On page 7, the authors state that "localized proliferation transformed the thin epithelium to a thickened outgrowth by generating an initial cell mass". It is not clear what is meant by "initial cell mass" as these are simple epithelia - there is only a single layer of cells in the endoderm and there is a single layer of cells in the ectoderm. These cells are undoubtedly changing shape to facilitate the buckling required to form a bud but the implication that cells are "amassing" is incorrect. Likewise, their use of the term "placode" (page 4) to describe the tentacle tip is contentious and does not add anything to this paper. The authors have not demonstrated an effect on the thickness of the tentacle tip epithelium in any of their treatments and thus there is no reason to invoke this term or defend this idea in the present study.

3. Lack of appropriate references

The "zig-zag" distribution of tentacles, the number of tentacles, and the organization of the tentacles along the directive axis was described by Stephenson in 1935 and the "short gastrodermal folds" (which are actually called "microcnemes") were described by both Stephenson 1935 and Crowell 1945. Indeed, nothing presented in Figure 1 (or the first paragraph of the results) is novel. Figures 2 and 3 are novel as they are the first attempt to characterize the temporal and spatial progression of tentacle addition in this animal and the authors have done a

very nice job with this description. The authors later discuss "hox dependent segmentation during embryonic development" (page 12) but fail to mention these observations were made previously by Finnerty et al 2004 and Matus et al 2006. Lastly, it should be noted that differentiating cnidocytes in the tentacle tips (page 8) were first described by Zenkert et al 2011 (not Babonis and Martindale, 2017); this should be cited appropriately.

4. Insufficient data quality

General - For all fluorescent images, the authors should provide the data in separate channels (in addition to composites) and provide high magnification images to demonstrate the specificity of each signal. For example, many images ostensibly show EdU-labeling in proliferating cells but some of the labels appear to be of the wrong shape/size to be nuclei (e.g., Figures 4F,5C). Furthermore, the authors suggest that cell proliferation increases in "both layers" in the presumptive tentacle bud (page 7), but their images appear to be max projections that show only ectoderm (Figure 4F-H). The number of EdU-labeled cells is used to assess effects on proliferation throughout the manuscript, but the authors report the number of EdU+ cells per "area" (Figure 4I,5D) and it is not clear that these areas are comparable. EdU values should be assayed relative to the total number of nuclei in the selected area not the area of the selection itself. The authors should also provide the raw nuclear counts to show that the number of DAPI-labeled nuclei is approximately the same, independent of the number of EdU+ nuclei, when comparing various regions.

Re: The role of TOR - It is not clear how the authors define some cells/tissues as "pRPS6-negative" when there seems to be a fair amount of red signal even in the rapamycin-treated animals (Figure 5C,F). This should be addressed when the authors provide higher quality image data (above). There is no indication of why different doses of rapamycin were used for the experiments presented in Figures 5C,F and while the authors report that rapamycin-treated polyps were "smaller" this is not demonstrated or quantified in Figure 5. They also claim that the general growth that precedes tentacle bud specification is TOR-dependent, but do not provide sagittal views showing pRPS6+ or EdU+ cells throughout the body column. Additionally, the use of an anti-ribosomal protein S6 kinase antibody without any validation is inappropriate. It is customary to perform a western blot the first time an antibody is used in a new system. At a minimum, the authors should confirm that *Nematostella* actually has an ortholog of this gene and provide evidence that it is expressed in the same tissue that is labeled by the antibody.

Re: The role of FGF - The authors say they induce premature stop codons in *FgfRb* using CRISPR/Cas9 but they do not show that this manipulation results in a loss of WT *FgfRb* mRNA (by in situ or by PCR/qPCR). Additionally, they suggest FGF signaling modulates the expression of the TOR pathway and localizes cell proliferation to the tentacle buds but they did not actually quantify the effects of *FgfRb* knockout on the number of EdU+ or pRPS6+ cells in their mutants. Thus, they cannot really support a relationship between proliferation, TOR signaling, and FGF, as presented. The in situ patterns in Figures 6, S10 and the supplemental movie do not match. Figure 6 shows endodermal expression of *FgfRb* and Figure S10 shows what may be ectodermal expression in the polyps but the quality of these images is too poor to really evaluate. This supplemental movie is largely uninterpretable as structures of very different size and shape all seem to be labeled with the *FgfRb* probe and yet the magnification is not high enough to determine if any of these structures are co-labeled with a nuclear marker.

In an effort to promote transparency in the peer review process, I choose to waive my anonymity.

We thank all reviewers for their comments and suggestions, which have significantly improved the manuscript. Please find below our point by point responses.

Reviewers' comments:

Reviewer #1 (Remarks to the Author):

Ikmi et al. describe an intriguing feature of postembryonic development in *Nematostella vectensis*, the addition of tentacles during the growth phase of the animals. They describe the pattern in which pairs of tentacles are added after the first four have formed in the transition to the primary polyp. This pattern shows little variation and does not follow a simple sequence of symmetrical additions. The authors then focus on the first pair of new tentacles and show that these require feeding for their formation and that locally elevated levels of phosphorylated RPS6 and an increase in cell proliferation precede their outgrowth. Confirming that pRPS6 reflects a role for TOR signalling, incubation with rapamycin reduces both pRPS6 and EdU labelling at the site of tentacle formation. Lastly, animals mutant for FGFRb fail to form tentacles after the four tentacle stage, despite being able to feed, suggesting that the elevated levels of FGFRb transcript at the sites of tentacle addition are related to a function of FGF signalling in directing tentacle formation in primary polyps. Overall, this is a very well executed study with beautiful illustrations of interesting and novel observations. Some aspects are probably not surprising, e.g. that growth requires feeding and that this requires TOR signalling. Still, it has to be shown and this has been done convincingly. Technically, there are only some points that I would like the authors to consider for improvements, all concerning the role of FGFRb.

1. The expression pattern in the colorimetric in situ in Fig S10 A and B (and in the Matus paper) resembles very much a potential in situ artefact of these late planula/early polyp stages. The stainings in S10C and D and in Fig 6 look more convincing, but they seem to be at a later stage. To support the proposed pre-metamorphic origin, would it be possible to test if these spots remain after shRNA injection? It is also not clear to me how the number of these "clusters" relate to the addition of the tentacles. The tentacles appear in pairs, but there seem to be more clusters of cells (also in the movie). This is a point that is not essential for the manuscript, but maybe the "pre-metamorphic" origin of the clusters is not so clear. The authors probably also see a different way to show that this particular staining in the in situ is specific.

To address this concern, we generated a novel transgenic line (*Fgfrb-eGFP*) that recapitulates the endogenous expression pattern of *Fgfrb*. We performed fluorescent *in situ* hybridization to visualize *Fgfrb* mRNA and immunostaining for α -eGFP in this line, and found an overlap between the endogenous and transgenic *Fgfrb* expression in those endodermal scattered cells in both larval and polyp stages, thus conforming the the pre-metamorphic origin of those clusters. Interestingly, by combining this transgenic that labels cell morphology with F-actin staining, we were able to assign the potential identity of these clusters to a sub-population of ring muscle cells. This new data is now shown in Figure 6 and Supplementary Figure 10, as well as described in the text page 10 paragraph 2.

2. Using FGFRb mutants for the analysis is great, but maybe the temporal control allowed by SU5402 treatment could be exploited to obtain more information about the temporal requirement of FGF signalling in this context. Obviously, the gene-specificity is lost, but maybe the comparison of the mutants with transient SU5402 treatments would be informative.

We agree with Reviewer 1 and performed the suggested SU5402 treatment experiment. As shown in the new Supplementary Figure 11, the spatial enrichment of pRPS6 staining and cell proliferation defining tentacle primordia in polyps was not detected in SU5402-treated animals compared to controls. This result is consistent with our findings in the *Fgfrb* mutant background. However, control and drug-treated polyps showed similar size increase following 5 days of feeding while the *Fgfrb* mutant exhibited a significant delay in organismal growth compared to their siblings. This temporal control of FGFR activity indicates that this signaling is dispensable for polyp growth while it is critical for polarized growth during feeding-dependent budding. This finding also suggests that the delayed growth in the *Fgfrb* mutants could be the result of the disruption of FGFRb activity during embryonic/larval development.

3. A cross-section of the FGFRb mutants showing the organization of the gastric cavity and mesenteries would be informative to understand the phenotype better. Also, it would be good to add more information about the development of the phenotype: when does the animals first differ from the sibs?

We agree and added new data that shows the organization of the gastric cavity and mesenteries in the *Fgfrb* mutants (see new panel E in Figure 7 and Supplementary figure 12). In brief, these mutants exhibited the expected eight body segments. However, they lack the clusters that premark tentacle primordia in polyps. They also show delayed metamorphosis and display reduced septal filaments and defects in longitudinal tentacle muscles compared to those of their siblings.

While these technical questions (and the minor points listed below) do not diminish my excitement for the data, I am more uncertain about the conceptual background and the implications of the manuscript. The authors embed their study in a context of “continuous developmental capacities” (introduction) and axial patterning (abstract). In the discussion, they mention developmental responses to changes in the environment (nutrition). There is a reference to regeneration, but the phenomenon under study is a regular developmental event that comes to an end at the 16 tentacle stage. To me the broader context remained very unclear. To justify publication at this level, the authors would need to make a more convincing point (evolutionary or developmentally) that the data are important beyond the regulation of one step of tentacle addition in this species.

To address this comment, we revised our text to highlight the two novel points of this manuscript. **The first point** is the relationship between embryonic and post-embryonic organogenesis. Our findings in *Nematostella* shows that the post-embryonic tentacle development is not simply a redeployment of embryonic mechanisms and these two processes rely on distinct patterning system. **The second point** is the mechanistic link between post-embryonic organogenesis and nutrition. We propose that the crosstalk between nutritional signaling (TOR) and developmental signaling (FGFRb) couples post-embryonic body patterning with food availability. Importantly, tentacle development in *Nematostella* does not end at 16T, but we stopped our analysis at 16-tentacle stage. To clarify this point, we added a new Supplementary Figure 2 showing that adult polyps can grow more than 18-tentacles (19T, 20T, 22T, 23T and 24T) when they are not regularly spawned. In addition, we provide new data in Supplementary Figure 3 showing that tentacle development can be triggered or arrested depending on the nutritional status of the environment. These findings highlight the environmental regulation of post-embryonic development in a long-lived animal and establish a novel experimental framework to study body patterning beyond embryogenesis and injury-induced regeneration.

Minor points:

1. On page 3 the authors mention that they here study adult cnidarians, but this is not the case.

We agree and updated the text.

2. The short folds are mentioned quite a while before there is a reference to them being shown in Fig S4. It might be helpful to have this reference earlier. How far do these short folds extend along the primary body axis?

As suggested by Reviewer 1, we moved this point to the first paragraph of the result section.

3. Is it unambiguous that Bodipy stains lipid droplets?

We confirmed the result of Bodipy staining with an independent method that stains lipid droplets which is the Oil Red O staining. See new Supplementary Figure 7.

4. What is the staining for the capsules?

Capsules were stained with DAPI. This is described in the Materials & Methods.

5. The description of the movie should be clearer. What are the two “episodes” of the movie (before and after rendering?).

We updated the movie and the description.

Reviewer #2 (Remarks to the Author):

Review on manuscript NCOMMS-19-17704

The authors in this manuscript describe the post-embryonic tentacle patterning in sea anemone *Nematostella*, showing that patterning is dependent on nutrient availability, induction of the mTOR pathway and activation of FGFR developmental signaling pathways.

The first part of the manuscript is a spatial and temporal description of the tentacles formation in *Nematostella* polyps (from 4 primary tentacles to up to 16-tentacles in the adult) and conclude on a stereotyped and reproducible spatial patterning system. The authors then show that development of a full set of tentacles is feeding-dependent, as unfed polyps arrests at the 4-tentacles stage, and also mTOR dependent, by using molecular markers for cell proliferation and TOR activity. Finally they show that the FGFR expression is essential for post-embryonic tentacle formation, by generating CRISPR/Cas9 mutants.

1- The authors suggest that mTOR-dependent growth activates the FGFR signaling pathway, which in turn generate a feed back loop to locally enhance mTOR pathway. It is not clear from the manuscript that there is a direct activation by mTOR of the FGFR expansion upon feeding. How is the FGFR pattern in presence of rapamycin?

We agree with Reviewer 2 that the initial manuscript was missing the analysis of *Fgfrb* expression in presence of rapamycin. In the revised manuscript, we performed this experiment and found that fed-polyps treated with 1 μ M Rapamycin do not show the expected expansion of *Fgfrb* expression observed in controls. This new data is shown in the panel F in Figure 6 and discussed in the text in page 11 paragraph 2. In addition, using the new *Fgfrb-eGFP* transgenic line and taking advantage of the different temporal dynamics between *Fgfrb* mRNA and eGFP expression, we found that the feeding-dependent expansion is nucleated around ring muscles expressing *Fgfrb*, associating these contractile cells with the pre-patterning of tentacle primordia.

Overall, the data is well presented and convincing, the manuscript is well written. The work is important for the field of patterning in response to environmental cues (here nutrients) as it sets the ground to further explore the molecular mechanisms regulated by mTOR and FGFR.

Minor remarks:

p3: "bauplan" should be in italics

It is now updated.

Figure 5F: rapamycin concentration is 3uM and not 1uM as indicated in text?

We corrected this concentration. It should be 1uM. In our experiments, we used those two concentrations and both showed identical effects.

Figure S2: title should implicitly indicate that these are unfed polyps

We updated the title. In this revised manuscript, Figure S2 became Figure S3.

--

Reviewer #3 (Remarks to the Author):

NCOMMS-19-17704

Summary

The authors examine the relationship between feeding, growth, and tentacle morphogenesis in the sea anemone *Nematostella vectensis*. They characterize the timing and sequence of tentacle formation in the juvenile polyp (after development of the four primary tentacles) and show that the onset of juvenile tentacle morphogenesis occurs only after the onset of feeding. Interestingly, they show that the initial process of primary tentacle formation is uncoupled from the process resulting in tentacles 5-16. Further, they suggest a role for the TOR pathway in both general growth and in juvenile tentacle morphogenesis and show that Rapamycin treatment is sufficient to block the formation of juvenile tentacle buds. They propose a role for FGF signaling in the development of specialized fields of cell proliferation and TOR signaling that demarcate the juvenile tentacle buds and further suggest that FGF is necessary to couple proliferation and TOR signaling during the general process of growth in these animals.

General Concerns

The description of the morphogenesis of tentacles 5-16 is thorough and while these data are novel, the phenomenon described is specific to *Nematostella vectensis* and as such, these results will not likely be of great importance to anyone outside of the *Nematostella* community. Furthermore, the claim that this study characterizes "lifelong axial patterning" is unfounded. In reality, the number of adult tentacles is largely invariant in this species (as noted by the authors and others before them) so the phenomenon they describe amounts to little more than a description of determinate post-embryonic growth. Although they have provided preliminary data to support a role for TOR and FGF signaling in the tentacle budding process, the data they provide are not of sufficient quality (or magnification) to assess the validity of these claims. Finally, throughout the manuscript, *the authors* use language that is imprecise and misleading and several key references have been excluded. Despite all of this, the amount of work that went into characterizing the spatial and temporal progression of tentacle morphogenesis is not trivial and these observations do constitute a

valuable contribution to the continued development of this emerging model system. I hope the authors can provide additional data to support the very interesting potential role of TOR and FgfRb in regulating juvenile tentacle morphogenesis. The data provided with this manuscript are, unfortunately, not of sufficient quality to substantiate their major claims. I have provided specific comments below intended to help improve the clarity and rigor of this study.

Specific Comments

1. "Lifelong axial patterning" is unfounded

It is not clear that tentacle morphogenesis is really feeding-dependent (abstract) or plastic (discussion).

Feeding is clearly required to induce the onset of tentacle morphogenesis but the authors have not demonstrated that animals will arrest at a number of tentacles below 16 if they are starved.

To address this concern, we added a new panels C and D in Supplementary Figure 3 data showing that post-embryonic tentacle development can be arrested at specific tentacle stages when food is not continuously provided.

Indeed, this study shows that starvation reduces the number of proliferating cells throughout the body, but not in the tentacle bud primordia (Figure 4), which explicitly argues against the claim that this process is feeding-dependent.

Our work shows that feeding is essential to trigger new post-embryonic tentacle development. When animals are starved after being exposed to food, the initiated tentacle development will proceed as our data show an enrichment of lipid droplets in tentacle primordia. This energy storage could serve as a buffering mechanism to complete tentacle development under unpredictable fluctuations of food supply. However, new tentacle buds will not develop until the animals are fed as supported by the data in Supplementary Figure 3.

Also, the term plasticity does not apply when the total number of adult tentacles is determined genetically; unless the authors can demonstrate that tentacles are lost when nutrients are limited or that nutrient limitation in the juvenile stage translates into a permanent change in adult tentacle number, then the use of the term plasticity is inappropriate.

There is no evidence favoring the hypothesis that the number of adult tentacles is solely determined genetically. So far, we cannot exclude the possibility that tentacle number could be defined by the interplay of genetic and environmental factors. In this case, 16T might represent a steady state that laboratory animals reach under the routine husbandry conditions. Supporting this point, we added a new Supplementary Figure 2 that shows adult polyps can grow more than 18-tentacles (19T, 20T, 22T, 23T and 24T) when they are fed and not regularly spawned, suggesting that there is a trade-off between resource allocation to reproduction and organogenesis in adult animals. The fact that tentacle number increases in adults in response to environmental changes is sufficient to call this phenotype plastic.

Similarly, the authors suggest (page 7) that tentacle budding "scales with body size" despite showing that budding is a binary response to feeding, independent of body size. As presented, there is no validity to this claim either.

To clarify this point, we updated the text. This experiment simply shows that reduced-sized polyps developed tentacle buds at smaller oral circumferences compared to full-sized

animals. This finding suggests that the initiation of new tentacle development in polyps does not depend on the ability of the animal to reach a specific body size, but the feeding-dependent tentacle development relies on a mechanism that scales in proportion to body size.

2. Imprecise language

On page 7, the authors state that “localized proliferation transformed the thin epithelium to a thickened outgrowth by generating an initial cell mass”. It is not clear what is meant by “initial cell mass” as these are simple epithelia - there is only a single layer of cells in the endoderm and there is a single layer of cells in the ectoderm. These cells are undoubtedly changing shape to facilitate the buckling required to form a bud but the implication that cells are “amassing” is incorrect.

We changed the word from mass to pseudostratification.

Likewise, their use of the term “placode” (page 4) to describe the tentacle tip is contentious and does not add anything to this paper. The authors have not demonstrated an effect on the thickness of the tentacle tip epithelium in any of their treatments and thus there is no reason to invoke this term or defend this idea in the present study.

We do not refer to tentacle tip as a placode. We use the term placode to describe the thickening and pseudostratification of tentacle primordia during embryonic/larval development as shown in our previous study (Fritz et al., 2013).

3. Lack of appropriate references

The “zig-zag” distribution of tentacles, the number of tentacles, and the organization of the tentacles along the directive axis was described by Stephenson in 1935 and the “short gastrodermal folds” (which are actually called “microcnemes”) were described by both Stephenson 1935 and Crowell 1945. Indeed, nothing presented in Figure 1 (or the first paragraph of the results) is novel.

The novelty of Figure 1 is that it shows a comparison between body organisation along the directive axis between primary and adult polyps. This has not been reported in previous literature, and is essential contextual information for a general scientific audience.

Figures 2 and 3 are novel as they are the first attempt to characterize the temporal and spatial progression of tentacle addition in this animal and the authors have done a very nice job with this description.

Thank you

The authors later discuss “hox dependent segmentation during embryonic development” (page 12) but fail to mention these observations were made previously by Finnerty et al 2004 and Matus et al 2006. Lastly, it should be noted that differentiating cnidocytes in the tentacle tips (page 8) were first described by Zenkert et al 2011 (not Babonis and Martindale, 2017); this should be cited appropriately.

Thank you- done.

4. Insufficient data quality

General - For all fluorescent images, the authors should provide the data in separate channels (in addition to composites) and provide high magnification images to demonstrate the specificity of each signal.

In this revised version, we updated the figures to provide separate channels and high magnification for the images.

For example, many images ostensibly show EdU-labeling in proliferating cells but some of the labels appear to be of the wrong shape/size to be nuclei (e.g., Figures 4F,5C).

In some experiments, we noticed that EdU was incorporated in the capsule of few stinging cells in the tentacle tip and pharyngeal regions. This mainly happened when the animals were starved or drug-treated. To clarify this point, we pointed out to those events in the updated figures.

Furthermore, the authors suggest that cell proliferation increases in “both layers” in the presumptive tentacle bud (page 7), but their images appear to be max projections that show only ectoderm (Figure 4F-H).

We added a new Supplementary Figure 6 to show cross section of both layers.

The number of EdU-labeled cells is used to assess effects on proliferation throughout the manuscript, but the authors report the number of EdU+ cells per “area” (Figure 4I,5D) and it is not clear that these areas are comparable. EdU values should be assayed relative to the total number of nuclei in the selected area not the area of the selection itself. The authors should also provide the raw nuclear counts to show that the number of DAPI-labeled nuclei is approximately the same, independent of the number of EdU+ nuclei, when comparing various regions.

Segmentation of nuclei in the oral pole to count cell is challenging due the cellular complexity of the pharynx region for which we could not provide a precise number. To overcome this challenge, we quantified the intensity of both Edu and Hoechst signals in a selected area of the oral pole spanning segments s3 and s7. We then plotted the correlation between the two intensities and used this as a proxy to quantify the distribution of proliferating cells with respect to nuclear density. This new analysis is now shown in Figure 4. The data strongly support our initial observation that there is a dramatic reduction of uniform cell proliferation the starved budded polyps while bud-localized cell proliferation is significantly less affected.

Re: The role of TOR - It is not clear how the authors define some cells/tissues as “pRPS6-negative” when there seems to be a fair amount of red signal even in the rapamycin-treated animals (Figure 5C,F). This should be addressed when the authors provide higher quality image data (above). There is no indication of why different doses of rapamycin were used for the experiments presented in Figures 5C,F and while the authors report that rapamycin-treated polyps were “smaller” this is not demonstrated or quantified in Figure 5.

To address this concern we now provide higher quality image data for pRPS6 staining in Figure 5. In addition, we added a new panel that show the distribution of the pRPS6 intensity at the oral pole in both control and Rap-treated polyps. The quantification of polyp size in the rapamycin experiment is shown in Supplementary Figure 5. In our experiments, we treated animals with 2 different concentrations of rapamycin and both showed identical effects. The concentration used in the experiments shown in Figures 5 and 6 is 1uM.

They also claim that the general growth that precedes tentacle bud specification is TOR-dependent, but do not provide sagittal views showing pRPS6+ or EdU+ cells throughout the body column.

We added a new Supplementary Figure 8 that shows sagittal views of pRPS6+ or EdU+ cells throughout the body column.

Additionally, the use of an anti-ribosomal protein S6 kinase antibody without any validation is inappropriate. It is customary to perform a western blot the first time an antibody is used in a new system. At a minimum, the authors should confirm that *Nematostella* actually has an ortholog of this gene and provide evidence that it is expressed in the same tissue that is labeled by the antibody.

This antibody is directed against a conserved phosphorylation motif in the 40S ribosome protein S6 that is present from yeast to human. Because of the high sequence similarity, this vertebrate antibody has been used non-animal species such as yeast (Yuan et al., JBC 2017). In this manuscript, we show the protein alignment in Supplementary figure 9. The relevant phosphorylation motif is well conserved in the 40S ribosome protein S6 ortholog in *Nematostella*. In addition, pRPS6 labelling is sensitive to both food/starvation and Rapamycin treatment, further validating that this staining reflects the phosphorylation state of RPS6.

Re: The role of FGF - The authors say they induce premature stop codons in *FgfRb* using CRISPR/Cas9 but they do not show that this manipulation results in a loss of WT *Frfrb* mRNA (by *in situ* or by PCR/qPCR).

Premature stop codons do not necessarily affect gene function at the transcriptional level unless they impact mRNA stability. The mutation we generated should directly impact the translation of the *FGFRb* mRNA and produce truncated protein products. However, we would expect to see a loss of *Frfrb* mRNA if *FGFRb* function is required for the survival of *Frfrb*-expressing cells. To test this possibility, we performed two independent experiments: 1. fluorescent *in situ* hybridization labelling of *Frfrb* mRNA in *Frfrb* mutants; and 2. analysis of the *Fgfrb-eGFP* reporter in *Frfrb* mutants. We observed that the discrete endodermal expression of *Fgfrb* is lost in the mutant but not the expression in other cell types, suggesting that *Fgfrb* function is essential for the development of the ring muscles. This new data is now shown in the Figure 7 and Supplementary Figure 12.

Additionally, they suggest FGF signaling modulates the expression of the TOR pathway and localizes cell proliferation to the tentacle buds but they did not actually quantify the effects of *FgfRb* knockout on the number of EdU+ or pRPS6+ cells in their mutants. Thus, they cannot really support a relationship between proliferation, TOR signaling, and FGF, as presented.

We added new panels in Figure 7 that show the requested quantifications.

The *in situ* patterns in Figures 6, S10 and the supplemental movie do not match. Figure 6 shows endodermal expression of *FgfRb* and Figure S10 shows what may be ectodermal expression in the polyps but the quality of these images is too poor to really evaluate. This supplemental movie is largely uninterpretable as structures of very different size and shape all seem to be labeled with the *FgfRb* probe and yet the magnification is not high enough to determine if any of these structures are co-labeled with a nuclear marker.

We have now added new figures and movies that more clearly show the expression pattern of *Frfrb* through both fluorescent *in situ* mRNA hybridization and our transgenic reporter line

(see the new Figure 6, Supplemental Figure 10 and Movies 1 & 2). This new Data should address the expressed concern.

In an effort to promote transparency in the peer review process, I choose to waive my anonymity.

Leslie S. Babonis, PhD

Reviewers' Comments:

Reviewer #1:

Remarks to the Author:

In the revised manuscript, Ikmi et al. made considerable efforts to address my concerns/comments. They chose, however, to provide only partial answers to some points. For the origin of the FGFRb expressing cell clusters it is now nicely shown that these expression domains are not an artefact. It remains unclear how the number of these clusters relates to the developing tentacles. In Fig S10D there are three such clusters visible and labelled, but a later stage (S10F) there are only two that label the new tentacle buds. The authors point out the latter observation but I still find it unclear whether the so-called "pre-metamorphic" clusters are really the origin of the new tentacle buds. The authors also did SU5402 treatments to show that the development of the tentacle buds requires FGF signalling after the first four tentacles are formed. This approach may also have been useful for earlier treatments to test whether the formation of the FGFRb clusters (and new tentacle buds) is affected before or during metamorphosis. There is now a better description of the phenotype of the FGFRb mutants at polyp stage, but how the phenotype develops over time, is not addressed.

These are non-essential points, but if the authors have the relevant data available, they could further strengthen the manuscript.

The authors have also provided a clearer description of the conceptual background and its relevance.

Other points:

line 113 micromeres should read microcnemes

line 121 – Please replace "adult organogenesis" with "tentacle development". While definitions of what can be called an organ are vague, the use of this term here is in my eyes an attempt to attract attention to animals that are typically said to lack organ-level organization.

Reviewer #3:

Remarks to the Author:

The authors failed to reference several of the seminal works on this topic (including Stephenson 1935 and Crowell 1946) but many of the observations they present have been published previously. *Nematostella* can have variable numbers of tentacles, which has been described, but the fact that wild populations typically have 16 (or fewer) tentacles suggests that the 20-, 22- and 24-tentacle animals the authors describe are aberrations that may not actually reflect the biology of this animal.

The authors continue to provide data that are of insufficient magnification and quality (and without proper counterstains) to evaluate their claims.

Figures 1-3 are fine, they have not changed from the first version.

Figure 4 The authors added two arrowheads pointing to "unspecific" EdU labeling in the first image in panel F but at the magnification they have provided, there is no way for any reader to discriminate "specific" from "unspecific" staining in these images. Thus, we are required to trust the authors' interpretation. The authors provided new panels showing quantification of the green signal in their images but have already pointed out that this label is not specific and seems to vary across treatments.

Figure 5 Panel C is identical in magnification and quality to the previous version and is still of insufficient quality to evaluate whether any of these labels (pRPS6 or EdU) are intracellular/nuclear. Panel D (or E?) claims to report RPS6 "intensity" (according to the rebuttal letter) but really these figures demonstrate EdU "intensity" (and I have already discussed my

concerns with this) and a count of the polyps the authors considered to be positive for pRPS6 expression.

Figure 6 Panel A is lovely, but panel B shows staining in some unidentified tissue at an oblique angle that is not interpretable, and panels C/D/E have no nuclear counterstain making it nearly impossible for anyone to know whether those cells are ectodermal or endodermal. Panel E seems to show ectodermal expression in the top panels (unfed) and endoderm in the bottom panels (fed) but it is not possible to be sure from the images provided.

Figure 7 Panel E shows two different planes from control and experimental animals so it is not possible for the reader to draw any reasonable conclusions about whether the "oral cell clusters" are still there or not. Furthermore, there is a fundamental difference in the staining represented in the top and bottom parts of panel F that is not mentioned. Either the antibody is responding differently to these tissues in different treatments or the authors have provided images from significantly different focal planes that reflect different parts of the cell/tissue biology of these animals.

Supplemental Figure 7 is impossible to interpret as there is no cellular level resolution.

Supplemental Figure 8 Both panels are presented in insufficient quality to see cellular level resolution. Furthermore, the images in the bottom of panel B claim to show a cross section but seems to be showing body wall ectoderm.

Supplemental Figure 12 Panel E is not presented in high enough magnification to evaluate their claims – are the cells of the (ectodermal) septal filaments gone or has the epithelium just become cuboidal instead of the normal columnar morphology?

Response to Referee #1

1. In the revised manuscript, Ikmi et al. made considerable efforts to address my concerns/comments. They chose, however, to provide only partial answers to some points.

For the origin of the FGFRb expressing cell clusters it is now nicely shown that these expression domains are not an artefact. It remains unclear how the number of these clusters relates to the developing tentacles. In Fig S10D there are three such clusters visible and labelled, but a later stage (S10F) there are only two that label the new tentacle buds.

To clarify this point, we added the following sentence in Page 15 paragraph 1:

“Consistent with the pattern of tentacle addition, the *Fgfrb*-positive ring muscles do not simultaneously engage in post-embryonic tentacle development, suggesting that there is an unknown mechanism that controls their deployment in time and space.”

2. The authors point out the latter observation but I still find it unclear whether the so-called “pre-metamorphic” clusters are really the origin of the new tentacle buds.

To provide a direct link between “pre-metamorphic” clusters and the oral clusters in polyps, we would have to establish a lineage tracing approach in *Nematostella* which goes beyond the scope of this work. In order to clarify this point, we added text on page 10 to read: “Interestingly, these *Fgfrb*-positive cell clusters were also found during larval development, indicating their potential pre-metamorphic origin (Figure S10).”

3. The authors also did SU5402 treatments to show that the development of the tentacle buds requires FGF signalling after the first four tentacles are formed. This approach may also have been useful for earlier treatments to test whether the formation of the FGFRb clusters (and new tentacle buds) is affected before or during metamorphosis.

We agree that it would be interesting to check whether the formation of FGFRb-positive cells is affected in either mutant or in SU5402-treated larvae in further studies of initial tentacle patterning. However, the current manuscript is focused on tentacle addition and the developmental role of the oral clusters in polyps. We therefore do not feel that adding these new experiments would enhance the main message or the impact of the work.

There is now a better description of the phenotype of the FGFRb mutants at polyp stage, but how the phenotype develops over time, is not addressed.

Since the FGFRb mutants are able to develop primary tentacles and our manuscript is centred on nutrient-dependent tentacle addition in polyps, we do not think that adding these new experiments would make a major change in the paper. Moreover, mutant and wild type larvae are morphologically indistinguishable until metamorphosis. To analyse how the mutant phenotype manifests over time, we would need to perform live imaging of individual embryos resulting from F1 heterozygous crosses followed by retroactive genotyping. We have attempted this approach previously, but meaningful live imaging of the metamorphic transition remains prohibitively challenging because of the high motility/contractility of the animals. There is no established protocol to achieve this experiment.

These are non-essential points, but if the authors have the relevant data available, they could further strengthen the manuscript.

We agree with and thank Referee 1 for these comments.

The authors have also provided a clearer description of the conceptual background and its relevance.

Thank you

Other points:

line 113 micromeres should read microcnemes

Done

line 121 – Please replace “adult organogenesis” with “tentacle development”. While definitions of what can be called an organ are vague, the use of this term here is in my eyes an attempt to attract attention to animals that are typically said to lack organ-level organization.

Done

Referee #3

1. The authors failed to reference several of the seminal works on this topic (including Stephenson 1935 and Crowell 1946) but many of the observations they present have been published previously.

This statement seems to imply that “many” of the novel observations we present in this work (post-embryonic tentacle development, nutrition-dependent regulation of tentacle addition, molecular regulation of tentacle development, CRISPR/Cas9-induced FGF mutant phenotypes, drug experiments, etc.) have been “published previously.” While seminal works from 75 years ago could indeed be cited for basic anatomical descriptions, it is unclear how this can be interpreted as a constructive comment (e.g. The authors should cite X and Y in paragraph Z). **Crowell 1946**, an initial description of north American specimens of the species, says literally nothing about tentacle development or addition. After carefully re-reading this paper to try to ascertain where exactly Referee 3 would like us to add a citation, it appears they could only be referring to the statements, “*Tentacles, 16, rarely less, in two cycles of eight each.*” and “*The relationship of the tentacles to the mesenteries is shown by Fig 2.*”, which is followed by a simplified hand drawing. Neither of these two statements reflects prior publication of ANY of the key observations on tentacle development reported in our paper. Still, for completeness, we cited Crowell 1946 in page 5 paragraph 1 (relationship of mesenteries to tentacles). Regarding **Stephenson 1935**, see below.

2. *Nematostella* can have variable numbers of tentacles, which has been described, but the fact that wild populations typically have 16 (or fewer) tentacles suggests that the 20-, 22- and 24-tentacle animals the authors describe are aberrations that may not actually reflect the biology of this animal.

Our entire description of the tentacle addition sequence is focused on events up to the 16-tentacle stage. Furthermore, in the main text of the paper we directly state:

“Nematostella polyyps can harbor a variable number of tentacles ranging from four to eighteen, but the common number in adulthood is sixteen^{4,10.}”

Somewhat comically, Reference 10 above is none other than **Stevenson 1935**, which the Referee erroneously claimed we “failed to cite.”

Regarding animals with 18+ tentacle numbers, we do not even describe them in the main text but do mention this in the supplementary information. Is the reviewer suggesting that we hide the fact that 18+ tentacle animals can be observed in the lab in the absence of spawning? There is no reasonable scientific basis to remove or conceal this data.

3. The authors continue to provide data that are of insufficient magnification and quality (and without proper counterstains) to evaluate their claims.

Figures 1-3 are fine, they have not changed from the first version.

Figure 4 The authors added two arrowheads pointing to “unspecific” EdU labeling in the first image in panel F but at the magnification they have provided, there is no way for any reader to discriminate “specific” from “unspecific” staining in these images. Thus, we are required to trust the authors’ interpretation. The authors provided new panels showing quantification of the green signal in their images but have already pointed out that this label is not specific and seems to vary across treatments.

EdU incorporation has been utilized in a large number of publications to visualize S-phase positive cells as a proxy for cell proliferation in *Nematostella* (e.g. Passamaneck and Martindale, 2012; Meyer et al., 2011; Fritz et al., 2013; Amiel et al., 2015 and many other papers).

Panels F and G are referred to in the manuscript to provide evidence for the changes in the cell proliferation pattern in unfed, fed and starved animals. In all these conditions, EdU labelling colocalises with DNA staining and only a few cells with unspecific labelling appear in the pharynx and tentacle tips of unfed animals. In our manuscript, we do not make any claim about these anatomical structures as the main focus of this work is the oral tissue that gives rise to tentacle buds.

To clarify this, we have now added a new panel in **Supp. Fig. 6** to avoid any confusion. This new panel shows high magnification views of the tissue of interest showing the colocalization of both signals.

4. Figure 5 Panel C is identical in magnification and quality to the previous version and is still of insufficient quality to evaluate whether any of these labels (pRPS6 or EdU) are intracellular/nuclear. Panel D (or E?) claims to report RPS6 “intensity” (according to the rebuttal letter) but really these figures demonstrate EdU “intensity” (and I have already discussed my concerns with this) and a count of the polyps the authors considered to be positive for pRPS6 expression.

Fig 5C shows the effect of Rap treatment on pRPS6 and EdU at the tissue scale in fed animals, which is the appropriate level of magnification required to visualise defects in tissue patterning. The data is also supported by quantifications for both Edu intensity (Panel D) and the number of animals showing pRPS6-positive and -negative buds (Panel E). In addition, we show pRPS6 intensity in wild type and mutant animals in Fig 7H. We do not feel that an additional level of analysis would extend the impact of the work.

5. Figure 6 Panel A is lovely, but panel B shows staining in some unidentified tissue at an oblique angle that is not interpretable, and panels C/D/E have no nuclear counterstain making it nearly impossible for anyone to know whether those cells are ectodermal or endodermal. Panel E seems to show ectodermal expression in the top panels (unfed) and endoderm in the bottom panels (fed) but it is not possible to be sure from the images provided.

To address this comment, we added new panels in **Supp. Fig. 10** that directly support the data shown in Fig. 6. Below is a detailed response:

Each of the analysed markers is described in detail with the appropriate nuclear staining when it is introduced for the first time (Figure 5 Panels A-B for pRPS6; Figure 6 Panels A-B for *FGFRb* expression). In order to make the panels that show the co-staining data simple and visually accessible to non-expert readers, we decided not to re-show nuclear staining although we have the data.

Figure 6A and B establish the dynamics of *FGFRb* mRNA expression in both tissues (stg.0, stg.1 and stg.2). Panel B is referred to in the manuscript to provide evidence for the expansion of *FGFRb* expression in the gastrodermis (“endoderm”, stage 1), followed by an activation of *FGFRb* expression in the epidermis (“ectoderm”; stage 2). It is not clear what the reviewer means by “unidentified tissue”. There are only inner and outer tissue layers (gastrodermis vs. epidermis), and these are clearly visible in the panel.

Panel C is referred to in the manuscript to show the morphology of *FGFRb*-expressing cells in unfed transgenic animals. In this context, *FGFRb*-positive cells are only present in the gastrodermis (stg. 0; see Panel A) and we provided staining for both mRNA and eGFP in Panel C. Panel D shows the relationship between *FGFRb* mRNA and *FGFRb*-eGFP in response to feeding (stg.1). To clarify this, we have included DNA staining in Supplemental Figure 10H.

Panel E shows how the pattern of pRPS6 staining correlates with *FGFRb* expression in response to feeding. To address all of these concerns, we have now updated **Fig. 6C-E** to clearly indicate the boundary between gastrodermis and epidermis based on the nuclear staining, which is shown in Supp. Fig. 10H-J.

6. Figure 7 Panel E shows two different planes from control and experimental animals so it is not possible for the reader to draw any reasonable conclusions about whether the “oral cell clusters” are still there or not.

Panel E does not show different planes. To clarify this point, we added two new **movies (S3 and S4)** showing z-stacks for each condition.

7. Furthermore, there is a fundamental difference in the staining represented in the top and bottom parts of panel F that is not mentioned. Either the antibody is responding differently to these tissues in different treatments or the authors have provided images from significantly different focal planes that reflect different parts of the cell/tissue biology of these animals.

Panel F is referred to in the manuscript to provide evidence for the lack of enrichment of Edu and pRPS6 in the oral tissue of fed *FGFRb* mutants. All the images are z-projections of similar planes. We do not make any claim about the scattered puncta observed in the mutants as we do not know their identity. To clarify this point, we added arrowheads that point out those puncta in **Fig 7F**.

8. Supplemental Figure 7 is impossible to interpret as there is no cellular level resolution.

In terms of word choice, “impossible” is quite a stretch in this case. Supplemental Figure 7 was added to the manuscript as requested by Referee #1 to extend the result of Bodipy staining in Figure 4 Panel J with an independent method that stains lipid droplets. In this context, the tissue scale resolution of the images was important to show the spatially patterned distribution of lipid droplets across the oral pole. We make no claim about cellular resolution when we state in the text: “Interestingly, an enrichment of lipid droplets was detected in bud primordia, visualized with BODIPY and Oil Red O staining (Figure 4J; Figure S7).”

Supplemental Figure 8 Both panels are presented in insufficient quality to see cellular level resolution. Furthermore, the images in the bottom of panel B claim to show a cross section but seems to be showing body wall ectoderm.

Supplemental Figure 8 shows side views of animals treated with Rapamycin and labelled with EdU and pRPS6. These side views include both surface views and cross-sections. This supplementary figure was added to complement the data shown in Figure 5 Panel C. The tissue scale resolution is required to appreciate the effect of Rapamycin on the rest of the body while still showing the location of tentacle primordia. To address this point we have now updated **Supp. Fig. 8B** to show a larger area of the gastric cavity as a clear illustration of a cross-section.

Supplemental Figure 12 Panel E is not presented in high enough magnification to evaluate their claims – are the cells of the (ectodermal) septal filaments gone or has the epithelium just become cuboidal instead of the normal columnar morphology?

Supplemental Figure 12 Panel E shows that mesenteries are present in the mutant animals, but with a reduced thickness of septal filaments. We do not make any claims about the cell morphology of this structure and it is not clear how adding an additional level of analysis would extend the impact of the work.